# Exogenous dsRNA triggers sequence-specific RNAi and fungal stress responses to control *Magnaporthe oryzae* in *Brachypodium distachyon*

Ying Zheng [1], Benjamin Moorlach [2], Desiree Jakobs-Schönwandt [2], Anant Patel [2], Chiara Pastacaldi [1], Stefan Jacob [3], Ana R. Sede [4], Manfred Heinlein [4], Minna M. Poranen [5], Karl-Heinz Kogel [4] ✉ & Maria Ladera Carmona [1] ✉

In vertebrates and plants, dsRNA plays crucial roles as PAMP and as a mediator of RNAi. How higher fungi respond to dsRNA is not known. We demonstrate that *Magnaporthe oryzae* (*Mo*), a globally significant crop pathogen, internalizes dsRNA across a broad size range of 21 to about 3000 bp. Incubation of fungal conidia with 10 ng/µL dsRNA, regardless of size or sequence, induced aberrant germ tube elongation, revealing a strong sequence-unspecific effect of dsRNA in this fungus. Accordingly, the synthetic dsRNA analogue poly(I:C) and dsRNA of various sizes and sequences elicited canonical fungal stress pathways, including nuclear accumulation of the stress marker mitogen-activated protein kinase Hog1p and production of ROS. Leaf application of dsRNA to the cereal model species *Brachypodium distachyon* suppressed the progression of leaf blast disease. Notably, the sequence-unspecific effect of dsRNA depends on higher doses, while pure sequence-specific effects were observed at low concentrations of dsRNA (< 0.03 ng/µL). The protective effects of dsRNA were further enhanced by maintaining a gap of at least seven days between dsRNA application and inoculation, and by stabilising the dsRNA in alginate-chitosan nanoparticles. Overall, our study opens up additional possibilities for the development and use of dsRNA pesticides in agriculture.

Double-stranded RNA (dsRNA) has been recognized as a promising tool for protecting plants against viruses, insects, fungal pathogens and oomycetes[1–4]. In most eukaryotes, dsRNA triggers RNA interference (RNAi), a gene regulatory pathway that promotes genome stabilization and defence against RNA viruses and viroids[5–7]. In crop protection, the RNAi mechanism can be exploited for pest and disease control, using either exogenous dsRNA (spray-induced gene silencing, SIGS) or host-expressed dsRNA (host-induced gene silencing, HIGS) to reduce essential gene activities of pests and pathogens, thereby reducing their virulence[8].

In addition to its role in RNAi, dsRNA is recognized as a pathogen-associated molecular pattern (PAMP) in vertebrates and plants[9–11]. This triggers pattern-triggered immunity (PTI), e.g. nonspecific defence responses, which potentially mitigate viral infections. In vertebrates, a set of dsRNA receptors has been identified which induce a multitude of cell-intrinsic and cell-extrinsic immune responses upon dsRNA recognition[12,13]. However, how fungi respond to dsRNA is largely unexplored. Successful plant protection with dsRNA is based on the assumption that most fungi have a functional RNAi system[14,15] and are therefore sensitive to dsRNA-mediated silencing of fungal

[1]Institute of Phytopathology, Research Centre for BioSystems, Land Use and Nutrition, Justus Liebig University Giessen, Heinrich-Buff-Ring 26, 35392 Giessen, Germany. [2]Fermentation and Formulation of Biologicals and Chemicals, Bielefeld Institute of Applied Materials Research, Bielefeld University of Applied Sciences, Interaktion 1, 33619 Bielefeld, Germany. [3]Institute of Biotechnology and Drug Research, Hanns-Dieter-Hüsch-Weg 17, 55128 Mainz, Germany. [4]Institut de biologie moléculaire des plantes, CNRS, Université de Strasbourg, 12 rue du Général Zimmer, 67084 Strasbourg, France. [5]Molecular and Integrative Biosciences Research Programme, Faculty of Biological and Environmental Sciences, University of Helsinki, Helsinki, Finland. ✉e-mail: karl-heinz.kogel@agrar.uni-giessen.de; maria.ladera-carmona@agrar.uni-giessen.de

target genes[16]. However, the question of whether dsRNA also has an innate immunity-inducing component in fungi remains unresolved[17].

In the present work, we address the question of how fungi respond to exogenous dsRNA, using *Magnaporthe oryzae* (*Mo*) as an experimental model. The hemibiotrophic fungus is the causal agent of rice blast disease and ranked as number one of the world's top ten plant pathogens with the highest scientific and economic importance[18]. *Mo* infects aerial and root tissues of a variety of Poaceae, including the grass model *Brachypodium distachyon* (*Bd*). RNAi proteins from the DICER-LIKE (DCL), ARGONAUTE (AGO) and RNA-DEPENDENT RNA POLYMERASE (RdRp) families are required for fungal virulence[19,20], and a wide range of host-derived microRNAs (miRNA) accumulate in infected tissues, including miRNA candidates predicted to sequence-specifically target fungal mRNAs coding for virulence or pathogenicity-related genes[20]. Consistent with this, *Mo* can internalise and process exogenous dsRNA from transgenic rice plants expressing these RNAs[21] as well as from dsRNA-treated rice leaves, resulting in sequence-specific silencing of corresponding fungal target genes[22].

Here, we have discovered that dsRNA is also immunogenic in addition to its RNAi-mediated gene silencing activity. dsRNA treatment partly inhibited the infection progress in a size and sequence-nonspecific manner. Consistent with this finding, dsRNA induced the fungal high osmolarity glycerol (HOG) pathway, a phosphorelay system, which is conserved across fungal species and relies on activation of a MAPK cascade that responds rapidly to different types of environmental changes such as osmotic stress, UV, high temperature, lipopolysaccharides and oxidative stress[23–25]. We show that these sequence-nonspecific PAMP-like effects are transient, whereas a pure sequence-specific RNAi activity of exogenous dsRNA is seen at lower dsRNA concentration, when a gap of 7 or 14 days between dsRNA application and fungal inoculation is applied and dsRNA is stabilized by chitosan-alginate nanoparticles. Overall, our data show that the effects of dsRNA on fungi and the diseases they cause can be diverse, opening up additional opportunities for the development and use of dsRNA pesticides in agriculture.

## Results

### *Magnaporthe oryzae* takes up exogenous dsRNAs of different lengths from liquid cultures and leaves

In order to determine whether the uptake of dsRNA by *Mo* is size-dependent, we analysed its ability to take up enzymatically synthetized fluorescent dsRNAs (10 ng/µL) ranging in length from 21 bp to 1775 bp (Suppl. Data 2). dsRNAs were incubated with conidia in liquid culture and imaged using confocal scanning microscopy (CSLM) at 24 h post-treatment (hpt). Fluorescent germ tubes (GTs) were identified in all treated samples (Fig. 1A). This analysis was extended by incubating conidia with dsRNA from bacteriophage phi6, which has a tripartite genome represented by three dsRNA fragments of which the smallest is 2948 bp in size. After additional treatment of the mycelium with micrococcal nuclease (MNase) to remove

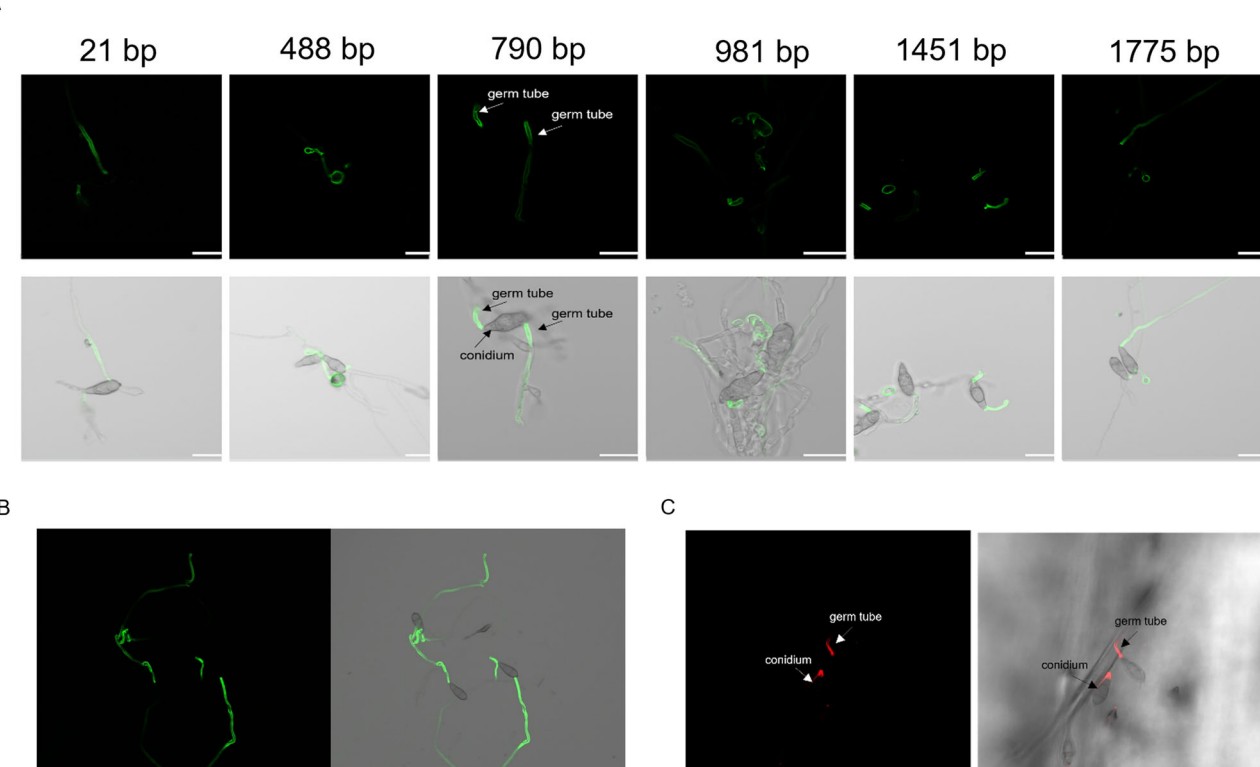

**Fig. 1 | Confocal laser scanning microscopy (CLSM) imaging of dsRNA uptake by *Magnaporthe oryzae* from liquid cultures and the leaf surface. A** Conidia ($10^4$ mL$^{-1}$) were incubated with 10 ng/µL dsRNA labeled with Fluorescein (21 bp dsRNA) or Alexa Fluor® 488 dye (longer dsRNAs; Suppl. Data 2) in 0.002% (v/v) Tween20 for 24 h at room temperature, before images were taken. Scale bar equals 20 µm. Upper row: AF488 and fluorescein imaging [λexcitation (nm): 501; λemission (nm): 591]; lower row: merge with bright field. **B** CLSM imaging of Phi6-dsRNA uptake. *Mo* conidia ($10^4$ mL$^{-1}$) were incubated with 10 ng/µL of Phi-dsRNAs (2948, 4063 and 7599 bp) labeled with Fluorescein as described under A. Before images were taken, mycelia were treated with dsRNA degrading MNase. Scale bar equals

50 µm. Left: AF488 and fluorescein imaging [λexcitation (nm): 501; λemission (nm): 591]; right: merge with bright field. **C** CLSM imaging of dsRNA uptake by *Mo* germ tubes from locally treated *Bd* leaves. Intact second youngest leaves of three-week-old *Bd* plants were first drop-treated with 20 µL drops containing 10 ng/µL Cy3-labeled 490 bp SHP-dsRNA in 0.002% (v/v) Tween20. After 3 h, the treated leaf areas were drop-inoculated with 10 µL drops containing 100 *Mo* conidia and imaged 48 h after inoculation. Scale bar equals 50 µm. Cy3 [λexcitation (nm): 565; λemission (nm): 626] signal left and merged with bright field, right. All dsRNA sequences are shown in Suppl. Data 2 and Suppl. Data 3.

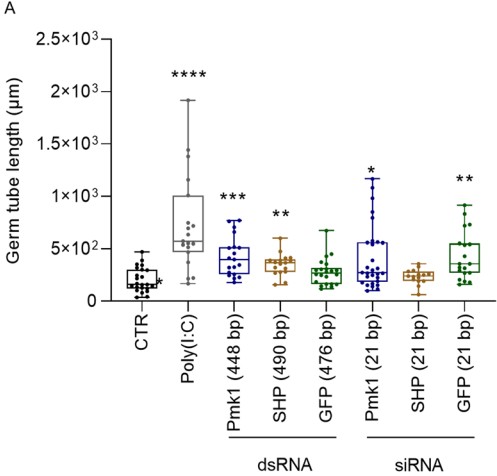
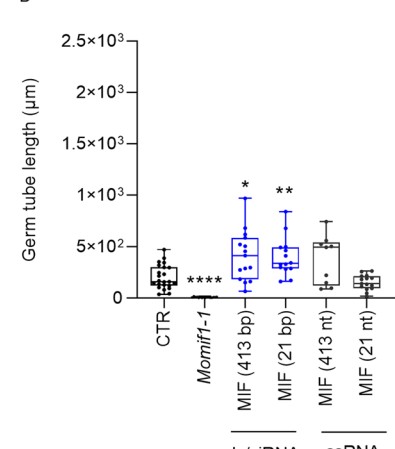

**Fig. 2 | *Magnaporthe oryzae* germ tube (GT) lengths in response to exogenous dsRNA.** *Mo* conidia (1000 in 200 μL) were incubated with 10 ng/μL of the indicated RNAs at room temperature for 24 h. Subsequently, conidia were germinated on coverslips for additional 24 h and the lengths were determined. **A** GT lengths of the wild-type *Mo* and (**B**) the MIF1 knockout mutant Δ*Momif1-1*. Images were taken with CLSM and GT length was analysed with ImageJ (and converted to micrometres). Data of three independent experiments were combined. Box plots represent average with standard deviation ($n \geq 9$). Statistical significance was assessed with Kruskal-Wallis test ($p \leq 0.05$) followed by Dunn's multiple comparisons test (A) (*: $p = 0.0166$; **: $p \leq 0.01$; ***: $p = 0.0007$; ****: $p \leq 0.0001$) and Welch ANOVA test ($p \leq 0.05$) with Dunnett test (**B**) (*: $p = 0.0215$; **: $p = 0.0027$; ****: $p \leq 0.0001$). Asterisks denote differences to the control group (CTR).

unabsorbed dsRNA, we confirmed that fluorescent Phi6-dsRNA is also taken up by the fungus. Taken together, these results show that the uptake of dsRNA by the fungal culture is independent of its size within the tested size range (Fig. 1B; Suppl. Data 2).

Next, we monitored the ability of *Mo* to take up exogenous dsRNA from dsRNA-treated *Bd*, an established host model for *Mo*[20]. The second youngest leaves of three-week-old *Bd* seedlings were treated with drops of Cy3-labeled dsRNA (490 bp SHP-dsRNA[26]; Suppl. Data 3) placed in the center of each leaf sheath. After 3 h, treated leaf areas were drop-inoculated with conidia. Confocal imaging at 48 hours post inoculation (hpi) confirmed that GTs can also take up fluorescent dsRNA from treated leaf areas (Fig. 1C).

### Exogenous dsRNA has a sequence-nonspecific effect on fungal development

In plants and mammals, long dsRNA ( > 30 bp) promotes sequence-unspecific PTI activities[9,27]. Here, we analysed the nature of dsRNA effects on a fungal pathogen by monitoring the impact of dsRNA on GT formation, since GT development is a good indicator of stress response in fungi[28]. To this end, we tested the dsRNA analogue polyinosinic:polycytidylic acid [poly(I:C)], which induces PTI responses in plants and animals[9,10], along with various *Mo*-targeting and non-targeting 21 bp siRNAs (21 bp duplex with overhangs) and long dsRNAs ( > 400 bp) derived from three different genes: the *Mo* MAP kinase *MoPkm1*, the *Sheath protein* (*SHP*) from the aphid *Sitobion avenae* (with no predicted target in *Mo*) and the *Green fluorescent protein* (*GFP*) (Suppl. Data 3). Most of the tested 21 bp duplexes and dsRNAs at concentrations of 10 ng/μL induced GT elongation in vitro as compared to the untreated control (Fig. 2A). This effect was dsRNA sequence-independent. The data suggest that exogenously applied dsRNA of different lengths can induce a sequence-independent response in a fungal culture.

Sequence-unspecific effects of dsRNA on fungal development was further characterized with the *Mo* knockout (KO) mutant Δ*Momif1-1*, in which the coding sequence of the *Macrophage Migration Inhibitor Factor 1* (*MoMIF1*) gene is deleted. As a result of this deletion, the Δ*Momif1-1* strain forms shorter GTs than the *Mo* wild-type[29]. To determine whether dsRNA-mediated gene silencing of *MoMIF1* results into a similar short GT phenotype, we treated germinating wild-type conidia with 10 ng/μL of MoMIF1-derived short (21 bp MIF1-siRNA) or long dsRNA (413 bp MIF1-dsRNA) and compared their GT length with the GTs of Δ*Momif1-1*. In agreement with our expectation[29], the GTs of Δ*Momif1-1* were significantly

shorter than the GTs of the untreated wild-type. In contrast, treatment with *MIF1*-derived siRNA and dsRNA induced longer rather than shorter GTs compared to the Δ*Momif1-1* strain or untreated wild-type strain, suggesting that treatment with these RNAs does not mimic the phenotype of the KO mutant (Fig. 2B; Welch ANOVA test, $p \leq 0.05$). We interpreted these results to suggest that the immunogenic effect of MIF-siRNA and MIF1-dsRNA at the given concentration of 10 ng/μL overrides the expected RNAi effect and masks the sequence-specific gene silencing activity and the associated inhibitory effect on GT development. Single-stranded (ss)RNAs identical in sequence to MIF1-dsRNA and MIF1-siRNA sequences had no significant effect on GT elongation (Fig. 2B), verifying that the observed immunogenic effects were triggered only by double-stranded molecules.

### Exogenous dsRNA induces ROS in fungal germ tubes

Reactive oxygen species (ROS) are involved in several important aspects of fungal development and pathogenesis, including the formation of conidia and infection structures[28,30,31]. Therefore, we speculated that exogenous dsRNA would have an inducing effect on the regulation of intercellular ROS in *Mo* seedlings. To address this question, *Mo* conidia were germinated and after 3 days, the hyphae were immersed into solutions containing siRNA or long dsRNA for 2 min and subsequently stained with $H_2$DCFDA, which reacts with intracellular $H_2O_2$ to form fluorescent dichlorofluorescein[32]. A fluorescent signal due to a strong ROS burst was triggered with the known stressor poly(I:C) and 10 ng/μL long dsRNAs, regardless of their sequences, but not with the equal concentration of siRNA (Fig. 3A). Furthermore, dose-response analysis in the range of 0.1 to 10 ng/μL showed that 5 ng/μL and higher concentrations of long dsRNA triggered $H_2O_2$ production, while the effect was weaker at 1 ng/μL and no effect was observed at 0.1 ng/μL (Fig. 3B).

### Exogenous dsRNA induces the canonical HOG stress pathway in *Magnaporthe oryzae*

In the ascomycete fungus *Trichoderma harzianum*, oxidative stress activates the stress marker *HOG1p*[23]. In order to gain more information on the fungal response to dsRNA, we addressed the question of whether dsRNA induces the high-osmolarity glycerol (HOG) stress pathway in *Mo*. This pathway plays a crucial role in the response of the fungus to various environmental stresses, including osmotic and oxidative stress, and also contributes to the regulation of fungal development, virulence and pathogenesis[33]. Activation of the pathway involves the migration of the HOG1 MAP kinase from the cytoplasm to the nucleus, where it triggers transcription factors that regulate

**Fig. 3 | Reactive oxygen species (ROS) burst detection by H$_2$DCFDA staining in hyphae of *Magnaporthe oryzae* in response to dsRNA treatment.** Three-day-old *Mo* liquid cultures were incubated with different RNA molecules for 2 min and subsequently stained for CSLM analysis. **A** Germinated conidia were treated with 500 ng/µL of poly(I:C) as positive stressor or 10 ng/µL of Pmk1-dsRNA and SHP-siRNA. **B** Dose-response effect of ROS production in germinating conidia after treatment with SHP-dsRNA. Fluorescence detection with AF488 [λexcitation (nm): 492; λemission (nm): 561]. Scale bar equals 50 µm.

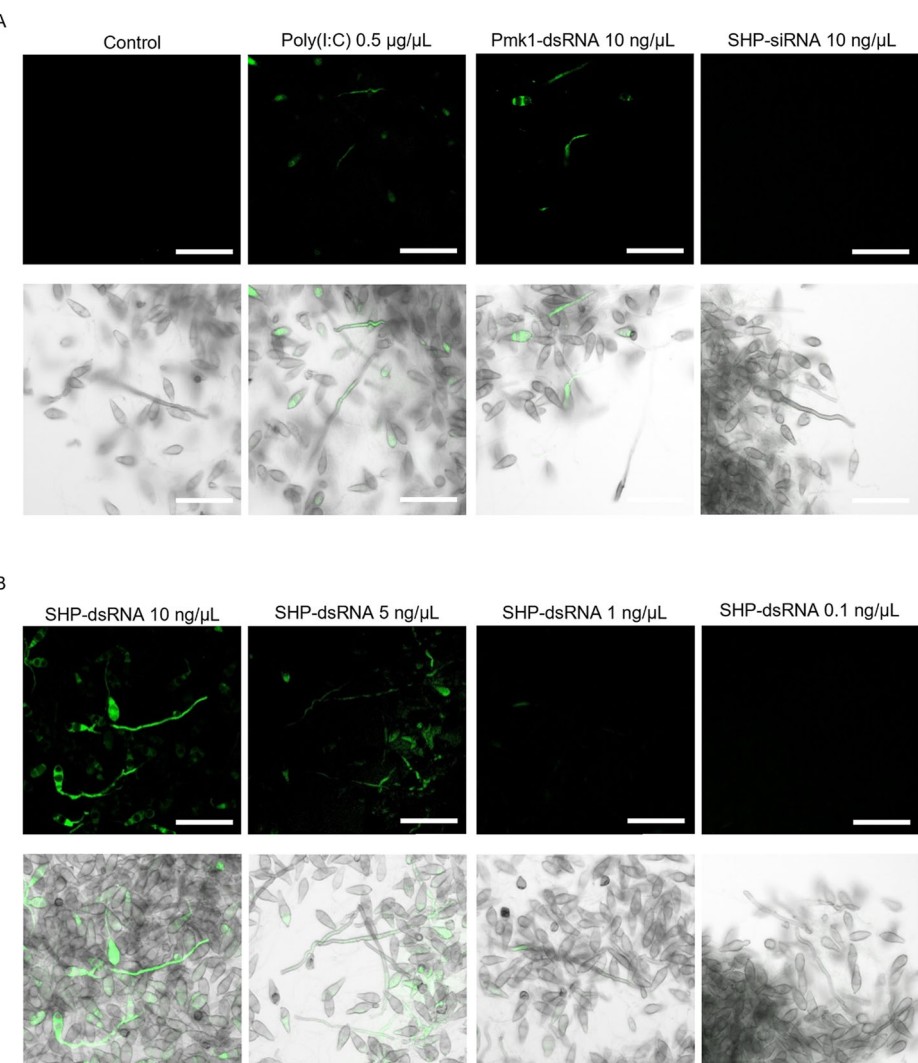

the expression of genes involved in the cellular stress response[34]. Conidia of a *Mo* reporter strain (HOG1::GFP) expressing a chimeric GFP-tagged *Mo* HOG protein (GFP-MoHog1p)[34] were incubated with various types of RNA. CSLM imaging showed that 20 ng/µL of poly(I:C), MIF1-dsRNA, SHP-dsRNAs or the corresponding siRNAs (MIF-siRNA, SHP-siRNA) induced the nuclear accumulation of the GFP-MoHog1p within 5 min of the treatment, regardless of whether they have a RNAi target in *Mo* (Fig. 4A, B). The response was similar to that observed in the presence of KCl (0.25 M), which was used as a positive control, while the aminoglycoside antibiotic geneticin, a protein biosynthesis inhibitor, was used as a negative control[34]. Instead, single-stranded (ss)RNAs such as 21-nt and 413-nt (sense) *MoMIF1*-specific ssRNAs and a 50 nt circular ssRNA (circRNA) with a sequence derived from *GFP* (Suppl. Data 3) did not induce migration of GFP-MoHog1p to the nucleus (Fig. S1). Of note, the positive control KCl induced only a transient nuclear accumulation of GFP-MoHog1p, while the effect of poly(I:C), dsRNAs and siRNAs persisted over 60 min, suggesting a strong response and activation of the stress pathway by different dsRNAs (Fig. 4A, B). Finally, we determined the minimum permissive dsRNA concentration in the range of 0.016 to 10 ng/µL that induces the HOG1 pathway of *Mo*. At dsRNA concentrations of 0.03 ng/µL and below, nuclear translocation of the chimeric protein was no longer observed (Fig. 4C). Overall, these data further confirm that dsRNA can induce stress responses in fungi in addition to the well-established RNAi-inducing activity.

## Exogenous dsRNA elicits sequence-specific and nonspecific effects on the infectivity of *Magnaporthe oryzae* on plants

Next, we tested the effect of exogenous dsRNA on *Mo* infection of *Bd* leaves. We used short and long dsRNAs targeting *MIF1* as previous work had shown that loss of *MIF1* strongly reduces the virulence of *Mo*[29]. Three-week-old *Bd* plants were sprayed with a suspension of conidia and 10 ng/µL MIF1-siRNA or MIF1-dsRNA. As a control, the plants were sprayed with a conidia suspension containing SHP-siRNA or SHP-dsRNA, which have no targets in *Mo*. Imaging and ImageJ-based measurement of the total necrotic leaf area at 5 days postinfection (dpi) showed that all dsRNAs and siRNAs significantly reduced necrotic blast symptoms (Fig. S2A, B). Consistently, the amount of fungus, as determined by the amount of fungal DNA relative to plant DNA, correlated with the occurrence of blast symptoms and was reduced in all leaves treated with dsRNA or siRNA (Fig. S2C; one sample *t*-test test, p ≤ 0.05). Taken together, these results show that both dsRNA and siRNA have a sequence-unspecific effect on fungal development on plant leaves.

To dissect sequence-specific and unspecific activities of dsRNA on fungal development, we performed experiments with additional *Mo*-targeting and non-targeting RNAs. We speculated that at lower concentrations the immunogenic activity of dsRNA might decrease while RNAi activity is maintained[35]. To this end, *Bd* plants were sprayed with a solution containing 10-fold lower concentration (1 ng/µL) of *Mo*-targeting (Pmk1-siRNA or Pmk1-dsRNA) or control (GFP-siRNA or GFP-dsRNAs) RNA. Consistent

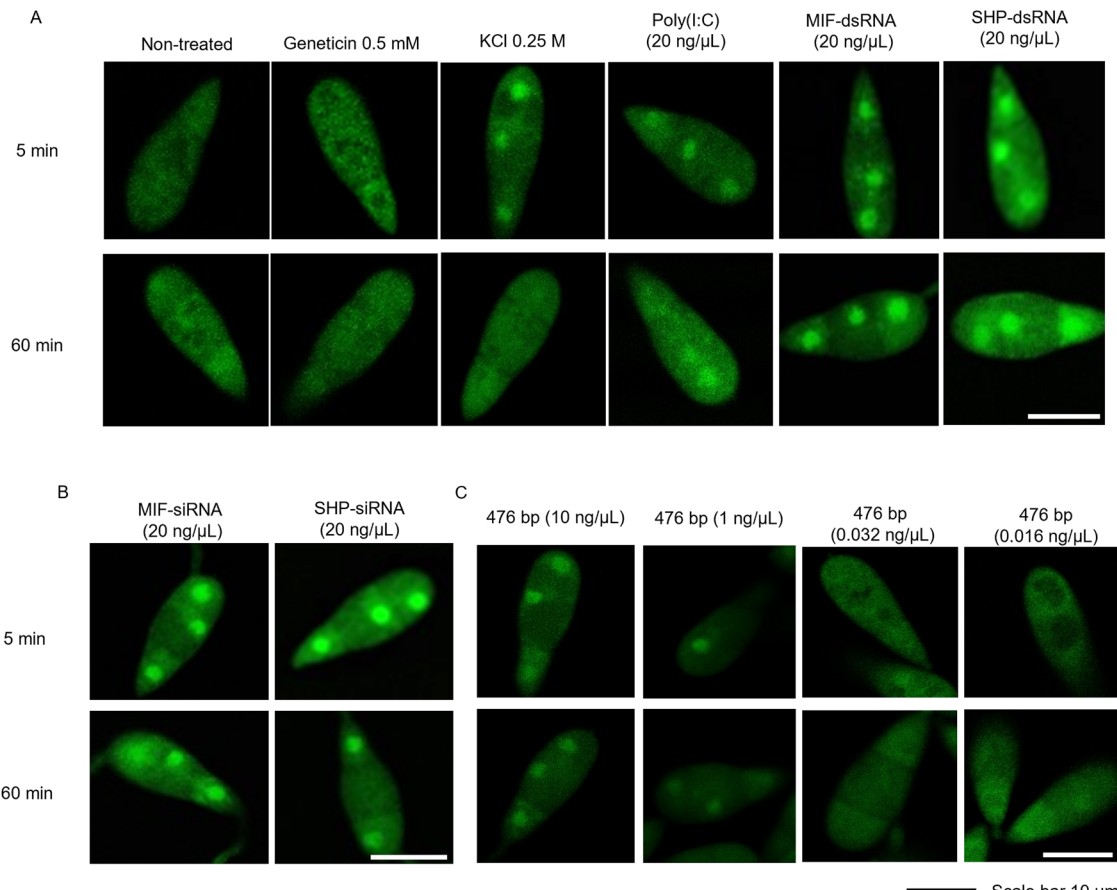

**Fig. 4 | Imaging the nuclear transfer of the chimeric GFP-MoHog1p in conidia of *Magnaporthe oryzae* in response to different stressors. A** Conidia (2000/100 μL) in 0.002% (v/v) Tween20 were incubated with 20 ng/μL of the indicated dsRNA. Potassium chloride (0.25 M) was used as positive stressor control, while geneticin was used as negative control. Images were taken with the AF488 laser at 5 and 60 min after treatment. **B** Conidia were incubated with 20 ng/μL of the indicated siRNA. **C** Dose-effect analysis of GFP-MoHog1p nuclear accumulation in *Mo* conidia treated with various amounts of GFP-dsRNA (476 bp). Scale bar equals 10 μm. AF488 [λexcitation (nm): 492; λemission (nm): 561].

with the above experiment, we found that *Mo*-targeting and non-targeting dsRNAs reduced the necrotic lesion area on *Bd* leaves (Fig. 5A). However, Pmk1-dsRNA and Pmk1-siRNA reduced infection symptoms significantly more than GFP-dsRNA or GFP-siRNA, which have no target in *Mo*, revealing a sequence-specific RNAi effect in addition to sequence-unspecific effects (Fig. 5B). Furthermore, quantification of the infection showed a significant reduction in the amount of fungus only in leaves treated with Pmk1-siRNA and Pmk1-dsRNA (one sample *t*-test, p ≤ 0.05) (Fig. 5C). Consistent with these data, microscopic analysis of detached *Bd* leaves inoculated with a mixture of *Mo* conidia and Pmk1-dsRNA (10 ng/μL) showed fewer appressoria formation at 2 days post-dsRNA treatment (dpt) compared to inoculated water controls (Fig. S3A, B). Moreover, Pmk1-dsRNA treatment reduced *Mo* penetration and formation of invasive hyphae thus explaining reduced leaf necrosis at 4 dpt (Fig. S3C).

### RNAi and the HOG stress pathway are triggered at different dsRNA concentrations in *Magnaporthe oryzae*

Next, we tested whether dsRNA at concentrations below the threshold for HOG stress pathway activation and ROS production would still be sufficient to trigger sequence-specific defence (RNAi) against *Mo* in infected *Bd* leaves. *Bd* leaves were treated with conidia and 0.03 ng/μL of Pmk1-siRNA/ Pmk1-dsRNA or GFP-siRNA/GFP-dsRNA. At 6 dpi, reduced infection symptoms were seen in leaves treated with Pmk1-siRNA and Pmk1-dsRNA but not with GFP-siRNA and GFP-dsRNAs (Fig. 6A; Fig. S4). These data suggest that exogenous Pmk1-dsRNA and Pmk1-siRNA can trigger RNAi

in *Mo* in a sequence-specific manner at concentrations where the dsRNA has no detectable effect on fungal stress-related ROS and HOG1 signalling.

To further confirm the sequence-specific RNAi-triggering activity of Pmk1-siRNA and Pmk1-dsRNAs, we measured *Pmk1* transcript levels in *Mo*-infected leaves. *Bd* plants were sprayed with conidia together with the respective RNAs and harvested at 12 hpi. At this time point, MoPmk1 is required for appressorium differentiation and thus infectious growth of *Mo*[36,37]. Consistent with our expectation, *MoPmk1* transcripts were reduced in infected plants treated with 0.03 ng/μL of Pmk1-siRNA or long Pmk1-dsRNA. In contrast, GFP-siRNA or GFP-dsRNA did not affect *MoPmk1* transcript levels (Fig. 6B).

### Development of alginate-chitosan nanoparticles for dsRNA formulation

RNAi activities of dsRNA can potentially be enhanced by using a dsRNA formulation in preventive treatments against fungal infections. To this end, we generated alginate-chitosan nanoparticles (NPs) with dsRNA cargo (Fig. 7A). The NPs form through electrostatic interactions of positive amine groups with negative phosphate and carboxyl groups from chitosan, dsRNA, and alginate, respectively. Analysis by dynamic light scattering (DLS) revealed a range of particle diameters for Pmk1-dsRNA-NPs of 229 nm, 104 nm, and 75 nm, depending on particle intensity, volume, and number distribution, respectively (Fig. 7B). Moreover, the Pmk1-dsRNA-NPs had a mean hydrodynamic diameter of 166 nm, which was not significantly different from empty NPs (Fig. S5A). Finally, the size and

**Fig. 5 | RNA-spray-mediated protection of *Brachypodium distachyon* plants against *Magnaporthe oryzae* infection.** (**A**) Infection symptoms on *Bd* leaves sprayed with a mixture of *Mo* conidia and dsRNA or siRNA. Intact three-week-old *Bd* plants were inoculated with a 0.002% (v/v) Tween20 containing conidia ($65 \times 10^3$ conidia mL$^{-1}$) and 1 ng/µL of siRNA or long dsRNA. Control plants were sprayed with 0.002% (v/v) Tween (Buffer) or with conidia in Tween solution (Untreated). For imaging, second youngest leaves were detached at 6 dpi and placed on 1% agar plates. Scale bar = 20 mm. **B** Relative size of necrotic area compared to the whole leaf area calculated with ImageJ. The results of three independent experiments are shown as box plots representing the average with standard deviation. Statistical significance was assessed with Kruskal-Wallis test ($p \leq 0.05$) and asterisks denote difference to the control group (CTR-untreated) according to Dunn's multiple comparisons test (*: $p = 0.046$; **: $p = 0.0086$; ****: $p \leq 0.0001$), while differences between GFP and Pmk1 specific RNAs were assessed with two-tailed Welch's *t*-test (**: $p = 0.0059$; ****: $p \leq 0.0001$). **C** Relative fungal growth determined by RT-qPCR comparing *Mo* housekeeping gene *MoGPD* with *Bd* housekeeping gene *BdUbi10*. The percentage of reduced fungal growth from three independent experiments was combined and represented as average with standard deviation. Statistical significance assessed with two-tailed *t*-test (**: $p = 0.0013$) and one sample *t*-test to the control (CTR) group (*: $p = 0.0207$; **: $p = 0.0096$).

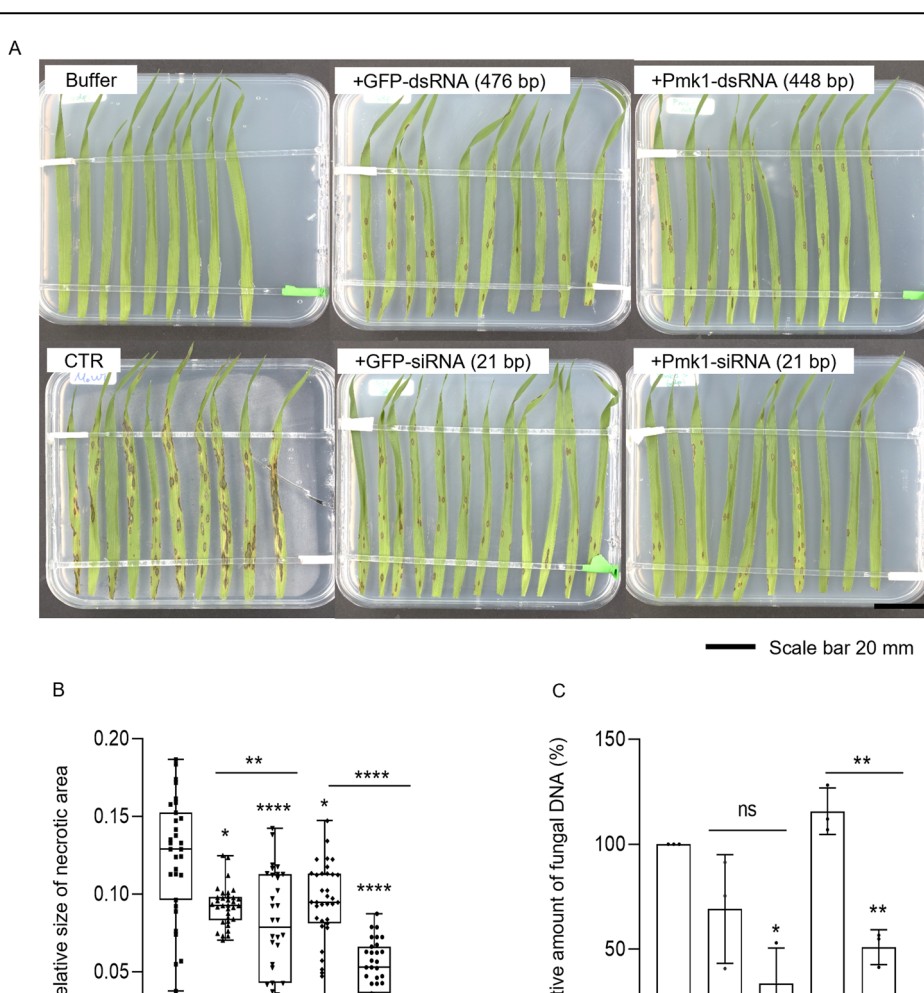

## Formulation prolongs the protective effect of dsRNA in *Brachypodium distachyon*

We compared the antifungal activity of naked Pmk1-dsRNA with formulated Pmk1-dsRNA-NP. Three-week-old *Bd* plants were first sprayed with a mixture of conidia and 1 ng/µL of dsRNA, dsRNA-NP, or empty NPs. Analysis of the necrotic leaf area at 5 dpi showed that unformulated and formulated Pmk1-dsRNA protected *Bd* plants to the same extent, whereas the empty NP did not provide any protection (Fig. 8A). Thus, dsRNA is released from NPs and can activate RNAi in the same extent as naked dsRNA.

Next, we designed an experiment, which better reflects the agronomic practise of a preventive treatment where a longer time interval between dsRNA application and inoculation could more precisely demonstrate the benefits of a dsRNA formulation. Plants were sprayed with dsRNA and inoculated with *Mo* conidia at 1 or 7 dpt. No difference between the plants treated with naked or formulated Pmk1-dsRNA was observed when a one-day gap (1-d-gap) was used between the treatment and the infection. However, when a 7-d gap was applied, the Pmk1-dsRNA-NP showed significantly stronger protective effect against *Mo* infection than the naked Pmk1-dsRNA (Mann-Whitney test, $p < 0.0001$) (Fig. 8B, C; Table 1).

## dsRNA-mediated RNAi is more robust towards control of *Magnaporthe oryzae* than PTI

In order to define if the protection provided by Pmk1-dsRNA against *Mo* infection (Fig. 8) was mediated by RNAi pathway or PTI, we performed an experiment in which we treated plants with *Mo*-specific and nonspecific dsRNA, and inoculated dsRNA-treated plants with *Mo* after 7 or 14 d. As nontargeting dsRNA, we used bacteriophage Phi6-specific dsRNA, which can be produced in high quantities using a cell-based production system we have developed earlier[38,39]. Blast symptoms and necrotic areas were reduced in *Bd* leaves treated with 1 ng/µL Pmk1-dsRNA-NPs when a 7 d gap was used between the treatment and *Mo* inoculation (Fig. 9A, B) and an even stronger effect was observed when a 14 d gap was used (Fig. 9D, E). However, unspecific dsRNA such as Phi6-dsRNA did not exert any effect regardless of its formulation (Fig. 9B, E). Thus, the protective effect observed at 7 or 14 dpt is mediated by RNAi, while nonspecific effects of dsRNAs are not sufficient to provide protection at these time points. Consistent with this, fungal quantification by quantitative reverse transcription polymerase chain reaction (RT-qPCR) showed that only Pmk1-dsRNA and Pmk1-dsRNA-NP treated plants had significantly reduced amounts of *Mo*, when plants were inoculated at 7 dpt (Fig. 9C). Furthermore, formulated Pmk1-dsRNA-NP had a protective effect also when inoculated at 14 dpt (Fig. 9F). Hence, non-formulated *Mo*-specific dsRNA could protect *Bd* plants

spherical morphology of NPs were confirmed by scanning electron microscopy (Fig. S5B, C).

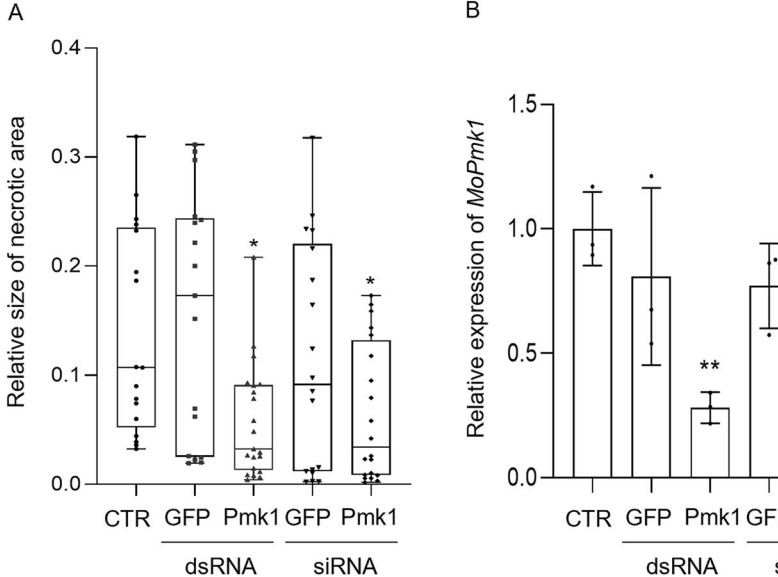

**Fig. 6 | Spray-induced protection of *Brachypodium distachyon* and gene silencing in *Magnaporthe oryzae*. A** Intact three-week-old *Bd* seedlings were sprayed with a solution containing conidia ($65 \times 10^3$ conidia mL$^{-1}$) and 0.03 ng/µL of the indicated siRNA and dsRNA. Infection symptoms were determined at 6 dpi and the relative size of the necrotic area was quantified using ImageJ. The results of three independent replicates were combined and box plots represent average with standard deviation. Statistical significance was assessed with Kruskal-Wallis test ($p \leq 0.05$) and asterisks denote difference to the control group according to Dunn's multiple comparisons test (* Pmk1-dsRNA: $p = 0.0279$; * Pmk1-siRNA: $p = 0.0267$.

**B** Silencing of the fungal *MoPmk1* gene in response to dsRNA treatment of *Brachypodium distachyon* leaves. Three-week-old *Bd* seedlings were sprayed with a mixture of *Mo* conidia and 0.03 ng/µL of fungal target-specific Pmk1-dsRNA, Pmk1-siRNA or *GFP*-dsRNA or -siRNA. Leaves were harvested at 12 hpi and analysed with RT-qPCR using *MoGPD* for normalization. Bars represent average of three experiments combined with standard deviation. Statistical significance was assessed with One-way ANOVA test ($p \leq 0.05$) and asterisks denote difference to the control group according to Dunnett test. (** Pmk1-dsRNA: $p = 0.0033$; ** Pmk1-siRNA: $p = 0.0024$). *Mo* indicates a non-treated control.

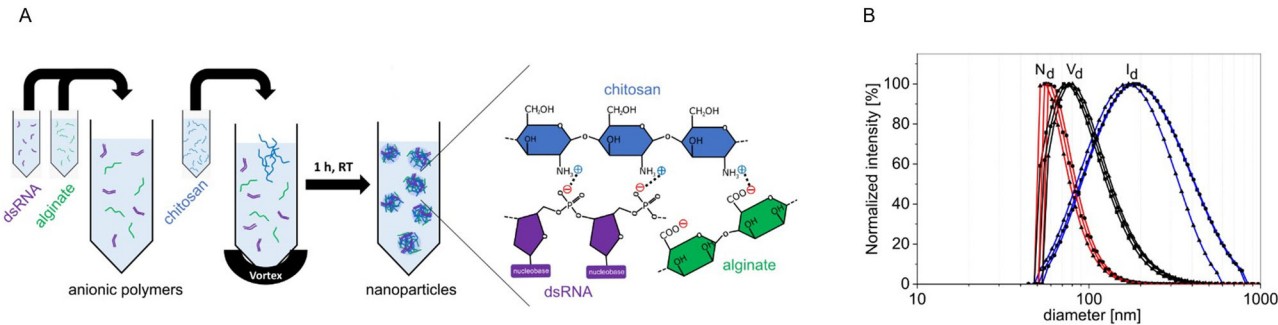

**Fig. 7 | Formulation of dsRNA in alginate-chitosan nanoparticles (NPs). A** Sketch of the formulation process of dsRNA-alginate-chitosan NPs. dsRNA and alginate are mixed in equal ratio to form an anionic solution. Chitosan is added to this solution, the mixture is vortexed and incubated at room temperature for 1 h. The NPs form through electrostatic interaction of positive amine groups with negative phosphate and carboxyl groups from chitosan, dsRNA, and alginate, respectively.

**B** Particle size distributions for Pmk1-dsRNA-NPs. Samples of Pmk1-dsRNA-alginate-chitosan NPs were characterized by dynamic light scattering (DLS). The peak of each distribution is the median of the distribution. Values are calculated by averaging the results from the three graphs shown in the figure. The normalized intensity ($I_d$), volume ($V_d$), and number distributions ($N_d$) of n = 3 samples (▪, ●, ▲, respectively) are on a logarithmic diameter scale.

from infection to a certain extent, but the protective effect was improved when the dsRNA was protected with the alginate-chitosan formulation. Furthermore, the results indicate that sequence-nonspecific dsRNA effects are not providing protection against *Mo* infection if there is a longer gap between dsRNA treatment and fungal inoculation, even when the dsRNA is formulated in NPs.

### dsRNA semi-systemically protects plants against *Magnaporthe oryzae*

Finally, we tested the semi-systemic effect of non-formulated vs. formulated Pmk1-dsRNA on *Mo* infections. While spraying dsRNAs, intact *Bd* leaves were covered either at the upper or at the lower part of the leaf, so that half of the leaf did not receive RNA directly through the surface. Following a 4-day gap, the whole plants were inoculated with conidia, and at 6 dpi, necroses

were analyzed in the leaf halves directly treated with dsRNA (locally-treated "LT") or indirectly treated (systemically-treated, "ST"), which would have received dsRNA via a possible systematic transport within the plant. As expected, we found that the area of necrotic lesion was significantly smaller in LT leaf halves as compared to leaves not treated with dsRNA. Moreover, smaller areas also were found in ST leaf halves but only when they were in acropetal direction to the directly treated LT leaf halves (Fig. 10). Remarkably, however, formulated dsRNA from the LT leaf also protected ST leaf halves in the basipetal direction (Fig. 10). These results confirm and extend previous data showing that dsRNA can move semi-systemically in a cereal leaf[40]. The majority of dsRNA moves in the acropetal direction, while only a small amount moves basipetally, consistent with the view that exogenous dsRNA moves via an apoplastic pathway[40,41].

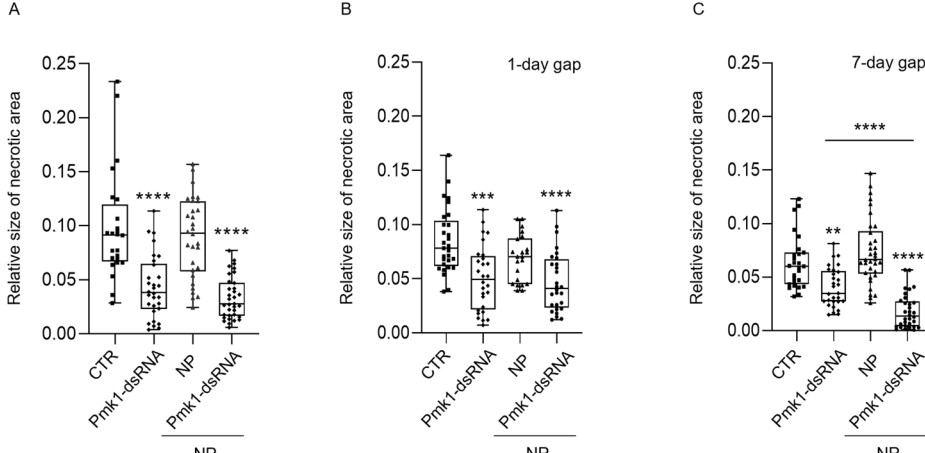

**Fig. 8 | The impact of dsRNA formulation on spray-mediated control of *Magnaporthe oryzae* on *Brachypodium distachyon* leaves.** Relative size of necrotic area on *Bd* leaves sprayed with *Mo* conidia and with naked Pmk1-dsRNA (1 ng/μL), chitosan-based Pmk1-dsRNA nanoparticles (NPs) or empty NPs. Control plants were sprayed only with conidia solution (CTR). Relative size of the necrotic area to the whole leaf was calculated at 5 dpi with ImageJ. (**A**) Simultaneous treatment with *Mo* conidia and naked or formulated dsRNA or empty NPs. **B** Sequential treatment with a time gap of one day between dsRNA/NP treatment and subsequent *Mo* inoculation. **C** Sequential treatment with a time gap of 7 days between dsRNA/NP treatment and subsequent *Mo* inoculation. The results of three independent repetitions were combined and box plots represent average with standard deviation. Statistical significance was assessed with Kruskal-Wallis test ($p \leq 0.05$) and asterisks denote difference to the control group according to Dunn's multiple comparisons test. Pairwise comparison in (**C**) was performed with Mann-Whitney test. (A, ****: $p \leq 0.0001$; B, ***: $p = 0.0003$; ****: $p \leq 0.0001$; C, **: $p = 0.0083$; ****: $p \leq 0.0001$).

## Discussion

In this work, we investigated the responses of the plant-pathogenic ascomycete fungus *Magnaporthe oryzae* to exogenous dsRNA. We use *Mo* as a proxy for invasive microbial pathogens and our results provide important mechanistic insights into fungal dsRNA responses. Knowledge on these responses is essential considering the future use of RNA-based antifungal strategies in agriculture and medicine. Importantly, our results show that dsRNA induce canonical fungal stress pathways in addition to RNAi. Strikingly, this stress response phenomenon was independent of the dsRNA sequence, occurred at high dsRNA concentrations and was rather transient compared to RNAi activity (Fig. S2; Fig. 5). From an agronomic point of view, a pesticide with multiple modes of action increases the resilience of crop protection strategies, can improve control efficiency, reduces the risk of resistance to an active ingredient and would thus contribute to sustainable agricultural practices.

### Table 1 | Disease severity in *Mo*-infected *Bd* leaves after treatment with naked and formulated dsRNA

| NO GAP | Disease severity | Avg. Necrotic area | St. Dev |
|---|---|---|---|
| Control | 100% | 0.0985 | 0.0507 |
| NP | 91% | 0.0895 | 0.0374 |
| Pmk1-dsRNA | 44% | 0.0433 **** | 0.0290 |
| Pmk1-dsRNA-NP | 34% | 0.0332 **** | 0.0193 |
| 1 DAY GAP | Disease severity | Avg. Necrotic area | St. Dev |
| Control | 100% | 0.0848 | 0.0300 |
| NP | 81% | 0.0688 | 0.0215 |
| Pmk1-dsRNA | 60% | 0.0508 *** | 0.0299 |
| Pmk1-dsRNA-NP | 56% | 0.0472 **** | 0.0276 |
| 7 DAY GAP | Disease severity | Avg. Necrotic area | St. Dev |
| Control | 100% | 0.0578 | 0.0244 |
| NP | 126% | 0.0726 | 0.0269 |
| Pmk1-dsRNA | 72% | 0.0414 ** | 0.0176 |
| Pmk1-dsRNA-NP | 30% | 0.0172 **** | 0.0179 |

**$p = 0.0083$; ***$p = 0.0003$; ****$p \leq 0.0001$

By using dsRNA of a broad size spectrum, we confirmed the ability of *Magnaporthe oryzae* to take up exogenous dsRNA (Fig. 1) as already previously reported for several fungi, such as *Fusarium graminearum*[42], *Verticillium longisporum*[43], *Sclerotinia sclerotiorum*[44], *Botrytis cinerea*, *Verticillium dahliae*, *Aspergillus niger*, and *Trichoderma virens*[45]. In contrast to reports on insects[2], our data show that the length of an exogenous dsRNA is not critical for fungal uptake. Instead, dsRNA was effective in a size range from 21 bp to 2948 bp (Fig. 1A), the latter representing the length of the smallest genomic fragment of the phi6 bacteriophage (Fig. 1B), while in some insects, the minimum length for dsRNA uptake is around 60 bp[46,47]. However, the minimum RNA length can vary depending on the insect species, and in some cases siRNAs of around 20 bp have been shown to be effective; this applies in particular to aphids[48].

Successful SIGS strategies, in which dsRNA applied to the leaf surface reduces fungal infection, have also been described for fungi, such as *Botrytis cinerea*[49], *Fusarium graminearum*[40], and *Magnaporthe oryzae*[22]. However, in light of these successful approaches to control a plant disease by spraying dsRNA, we were interested in investigating the activity of dsRNA on fungi in more detail. Whether SIGS experiments work because of the RNAi effect of dsRNA or because dsRNA has an effect on the innate immune system of a plant and/or a fungus has not yet been satisfactorily investigated. Moreover, the question of whether sprayed naked RNA is effectively taken up by intact, non-detached leaves is still controversial[50]. By addressing the latter question, we designed an experiment in which we tested RNA uptake by the fungus from sprayed surfaces (local uptake). The design of these experiments followed those applied to *Fusarium*[40] and *Magnaporthe*[22], with the important modification that we used intact plants (rather than detached leaves) for our experiments. *Mo* GTs accumulated fluorescence from these treated leaves, indicative of dsRNA uptake from treated leaf surfaces (Fig. 1C). Growing hyphae have thinner cell wall allowing exogenous dsRNA uptake, probably via clathrin-mediated endocytosis as shown in *S. sclerotiorum*[44]. Conidia have thicker cell walls and cannot hardly take up dsRNA. Consistent with this finding, we found a fluorescent signal in the germinating germ tube, not in the conidia (Fig. 1). We also tested whether naked or formulated dsRNA would confer systemic plant protection. Several reports showed naked dsRNA associated with xylem transport[41,51], which would follow acropetal direction. Indeed, we found acropetal protection in plants treated with naked dsRNA, while basipetal protection became only detectable when the dsRNA was formulated in NP (Fig. 10). We concluded that dsRNA can

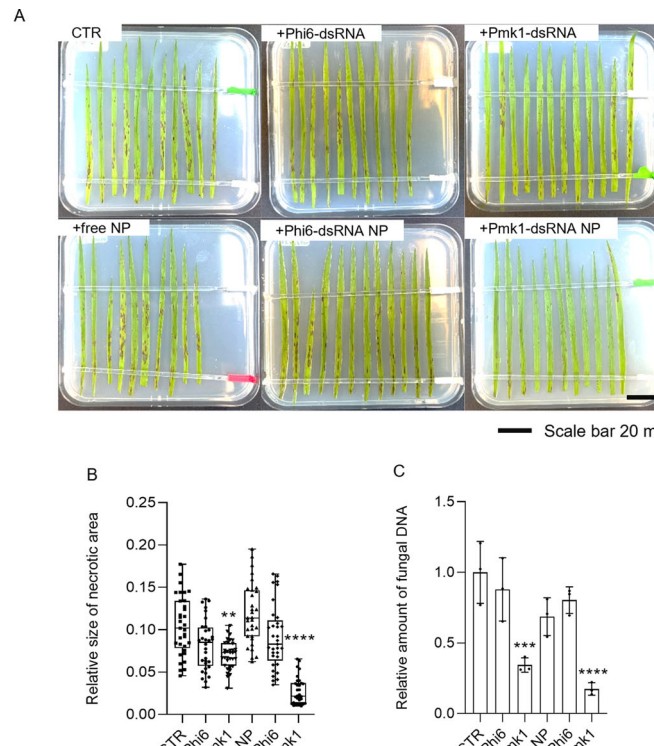

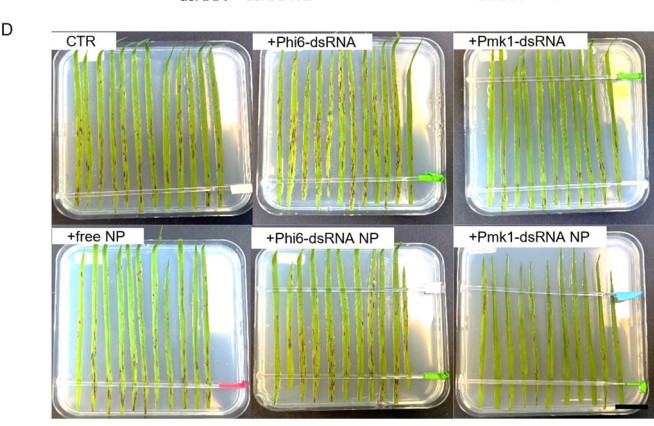

— Scale bar 20 mm

**Fig. 9 | Spray-mediated control of *Magnaporthe oryzae* on *Brachypodium distachyon* leaves.** Intact *Bd* plants were sprayed with dsRNA or NPs (1 ng/μL) and infected with *Mo* conidia at 7 (**A–C**) or 14 dpt (**D–F**). Plants were treated with empty NPs, naked Phi6-dsRNA or Pmk1-dsRNA, or formulated Phi6-dsRNA (Phi6-dsRNA NP) or Pmk1-dsRNA (Pmk1-dsRNA NP). Control plants were sprayed only with conidia suspensions (CTR). The results of three independent repetitions were combined. (A and D) Infection symptoms on *Bd* leaves. One representative picture from three independent experiments is shown. Scale bar = 20 mm. **B** and **E** Quantification of necrotic area on *Mo*-infected *Bd* plants. The relative size of the necrotic area to the whole leaf was quantified with ImageJ at 6 dpi. Box plots represent average with standard deviation. Statistical significance was assessed with Kruskal-Wallis test (p ≤ 0.05) and asterisks denote difference to the control group according to Dunn's multiple comparisons test (**: p = 0.0026; ***: p = 0.0001; ****: p ≤ 0.0001). **C** and **F** Relative fungal growth determined from harvested leaves by RT-qPCR based analysis of *Mo* housekeeping gene *MoGPD* expression. *Bd* housekeeping gene *BdUbi10* was used for normalization. Bars represent average with standard deviation from three independent repetitions combined. Statistical significance was assessed with One-way ANOVA (p ≤ 0.05) and asterisks denote difference to the control group according to Dunnett's multiple comparisons test. (*: p = 0.0187; ***: p = 0.0006; ****: p ≤ 0.0001).

of the knowledge that eukaryotes have evolved defence systems against viruses based on the recognition and degradation of dsRNA. Consistent with this view, dsRNA behaves like a PAMP in plants[1,9]. Furthermore, in vertebrates, apart from its RNAi-triggering activity, dsRNA is sensed by a number of innate immune receptors and elicits a variety of cell-intrinsic and cell-extrinsic immune responses upon recognition[52,53]. These RNA sensors include RIG-I- like receptors (RLRs), protein kinase R (PKR), oligoadenylate synthases (OASes), Toll-like receptors (TLRs) and NOD-, LRR- and pyrin domain- containing 1 (NLRP1)[13]. It was therefore plausible that, in addition to plants and vertebrates, fungi also exhibit a differential response to dsRNA that is not restricted to the canonical RNAi mechanism.

Consistent with our finding that dsRNA can act non-specifically on *Mo*, the fungal *MoHOG1p::GFP* reporter line, which was originally designed to monitor osmotic stress and fungicide effects in vivo[34], responded in a dose-dependent manner to dsRNA, but not to ssRNA having corresponding sequence (Fig. 4; Fig. S1). The HOG pathway and HOG1 phosphorylation is conserved across fungal species and relies on a MAPK cascade activation rapidly overcoming osmotic stress. The pathway is also associated with other types of stresses in different fungi, including UV, high temperature, lipopolysaccharides, and oxidative stress[23–25]. Notably, beside linear ssRNA, circular ssRNA (circRNA) also failed to induce the HOG pathway in *Mo*. This finding is consistent with a recent report showing that this type of RNA does not induce innate immune activation or antiviral responses in mammalian cells[54]. Of note, 20 ng/μL of dsRNA or sRNA is equivalent to 61.8 nM and 1.44 μM, respectively, a much lower molarity than the molarity of KCl (0.25 M) used as a positive control for HOG1 pathway induction, clearly indicating that the induction of HOG1 pathway by dsRNA is not caused by osmolarity (Fig. 4).

In search of a more mechanistic explanation for the unspecific effects of dsRNA on *Mo*, we also tested the possibility that ROS may play a role in the fungal response to dsRNA and siRNA duplexes. Fungi can produce ROS and accumulate intracellular $Ca^{2+}$ as the first response to stress. Injured hyphae in *Trichoderma atroviride* were stained with $H_2DFCA$, indicating a ROS burst after mechanical stress[32]. ROS formation relies on NADPH-dependent oxidases (NOX), and these enzymes have been found to participate in fungal differentiation as well[55]. ROS activate the HOG1 pathway in *Trichoderma harzianum, Saccharomyces species* and *Candida albicans*[23,56,57], while ROS is needed for proper GT elongation in *Puccinia striiformis*[28] where $H_2DCFDA$ staining revealed ROS accumulation in the germinating hyphae. Notably, high concentrations of ROS are detrimental for the *Mo* fungus[22]. To this end, we carried out a staining assay for $H_2O_2$ based on the dye $H_2DCFDA$. In line with previous experiments, long dsRNAs, including poly(I:C), caused a very early ROS burst in *Mo* (Fig. 3). Furthermore, longer dsRNAs exerted

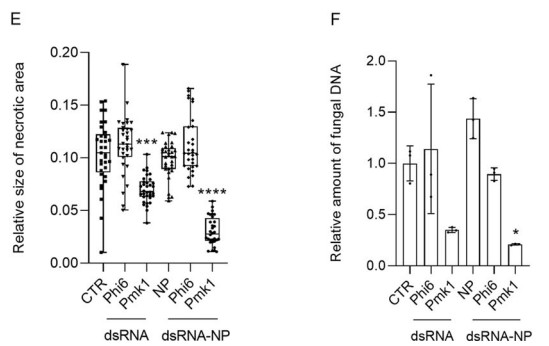

move semi-systemically in a cereal leaf. The majority of dsRNA moves in the acropetal direction, while only a small amount moves basipetally, consistent with the view that applied dsRNA distributes in the apoplast[40,41].

The effect of sRNA duplexes and dsRNA, including non-targeted dsRNAs homologous to *GFP* and *SHP*, on GT length was independent of RNA sequence and thus non-specific (Fig. 2). These results are reminiscent

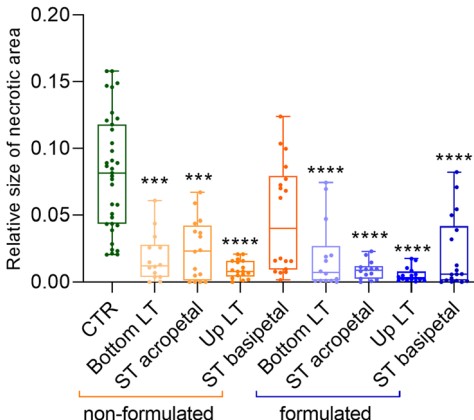

**Fig. 10 | Semi-systemic RNA-spray-mediated protection of *Bd* leaves.** Intact plants were sprayed with 1 ng/μL of dsRNA or dsRNA-NP at their upper or lower half of the leaf, while the other part was covered. After 4 days, complete plants were spray-inoculated with conidia. Analysis of necrotic area was performed at 6 dpi. The results from three independent repetitions were combined. Size of the necrotic area relative to the whole leaf area calculated with ImageJ. Box plots represent average with standard deviation. Statistical significance was assessed with Kruskal-Wallis test ($p ≤ 0.0001$) and asterisks denote difference to the control group according to Dunn's multiple comparisons test. CTR control leaves sprayed only with *Mo* conidia, LT locally treated leaves in their upper o bottom half, ST systemically treated in acropetal or basipetal direction. (*** bottom LT: $p = 0.0002$; *** ST acropetal: $p = 0.0003$; ****: $p ≤ 0.0001$).

a stronger effect compared to shorter ones, but this might be due to the 5' phosphorylation state of the in vitro produced long dsRNA. Enzymatically produced dsRNA contains 5' triphosphate group, that is absent in the chemically synthesized dsRNA like the siRNAs used in this study. In plants, longer dsRNA, such as poly(I:C), but not siRNA, triggered MAPK activation, a hallmark of PTI[9]. Altogether, our data suggest that also in *Mo*, longer dsRNA molecules activate stress signalling pathways in a more efficient way than shorter sequences (Fig. 2, Fig. 3). Further research is needed to determine whether this is a general response of fungi to dsRNA. However, the signalling response occurs very fast and transiently, in accordance to previous assays in other fungi where ROS were detected within minutes of injury[32]. We hypothesized that ROS burst triggered by dsRNA (Fig. 3) impacted germ tube development (Fig. 2) and Hog1p migration to the nucleus (Fig. 4). Microscopic analysis of *in planta* dsRNA treatments and *Mo* infection showed that dsRNA-treated conidia produced more aerial hyphae and reduced plant penetration (Fig. S3). Exogenous dsRNA perception might take place by any of the components upstream the MoHog1p signalling cascade, having a dual role, responding to osmotic stress but also to dsRNA. This is not surprising, since fungal cells might recognize the dsRNA molecules as a putative invader, i.e. a mycovirus, reprogramming a set of genes to respond to the infection. In the fungus *Malasezzia* that is infected by a mycovirus, genes involved in stress response, translation and even phosphorylation processes were upregulated, while genes involved in cell division and metabolism were down-regulated[58].

Pest management through formulated dsRNA has become a widely discussed and promising new tool for crop production[59]. Formulations can prolong the stability of dsRNA, and also enhance its cellular uptake, thereby improving efficacy[60,61]. Chitosan has been widely used for clinical purposes and in agriculture as antimicrobial agent, and also to alleviate plant stress[62,63]. The US Food and Drug Administration (FDA) conferred the GRAS status (Generally Regarded As Safe) to chitosan (GRAS Notice No. GRN 000997; equivalent to basic substances in the EU). We could demonstrate improved efficiency for dsRNA formulated with alginate-chitosan-nanoparticles (NPs) (Figs. 8, 9). Using this formulation, we were able to show in an experimental setup with a long-time-interval between dsRNA application

and inoculation that the sequence-unspecific effect of dsRNA is lost over time and thus proves to be transient compared to the sustained RNAi activity.

It is well known that exogenous dsRNA induces canonical PTI responses in plants, leading to enhanced protection against viral infection[9], while our work have focused on dsRNA-triggered nonspecific effects in the fungus, which are less well studied. Either way, both mechanisms would contribute to dsRNA-mediated protection against phytopathogenic fungi. All in all, our data open up further questions on the application of dsRNA in plant protection. The biological effects of dsRNA are more diverse than assumed in the first reports on its use as a nature-based pesticide, so that further research on the mode of action and practical application strategies, including information on the amount of active ingredient under field conditions, is urgently needed.

## Methods

### Plant and fungal growth conditions and inoculation protocols
The fungal strain used in this study was *Magnaporthe oryzae* (*Mo*) 70-15 (Fungal Genetics Stock Center, Kansas State University, Manhattan, USA). The strain was grown at 26°C on oatmeal agar (OMA; 50 mg/L Oatmeal, 10 mg/L agar). Conidia were harvested with 0.002% (v/v) Tween20 from 14-day-old *Mo* cultures grown on OMA plates and filtered through two layers of miracloth (Merck KGaA, Germany). Conidia concentration was calculated using 20 μL of the spore solution in a hemocytometer, and the final concentration was adjusted with 0.002% (v/v) Tween20.

*Brachypodium distachyon* cv. Bd21-3 was cultivated in soil (Fruhstorfer Erde Typ T, Vechta, Germany) in a growth chamber at 18°C/14°C (16 h light/8 h dark cycle) with 60% relative humidity and a photoperiod of 240 μmol m$^{-2}$ s$^{-1}$ photon flux density. Each pot contained 6 plants and two pots represented one experimental repetition. For SIGS assays, three-week-old intact plants were infected by leaf-spray inoculation with a suspension of $65 × 10^3$ mL$^{-1}$ conidia in 0.002% (v/v) Tween20 and 1 mL of conidia solution was sprayed in two pots (one experimental repetition) until water run-off. Buffer control plants were mock-inoculated with 0.002% (v/v) Tween20 or control plants were inoculated with *Mo* conidia only. Alternatively, dsRNA was mixed with *Mo* conidia and sprayed simultaneously. For gap experiments, plants were first sprayed with dsRNA in 0.002% (v/v) Tween20 and then with *Mo*. For semi-systemic protection, the lower half or upper half of the leaf was covered with aluminium foil when the dsRNA was sprayed. A detailed description of each experiment is given in the figure legends. For quantification of disease symptoms, each second youngest leaf was cut at 5 or 6 dpi, placed on 1% agar plates, and unfolded with the help of a plastic stick for better image acquisition. Then relative size of the necrotic area compared to the whole leaf area was quantified using ImageJ free software (https://imagej.net/ij/). The results of three independent experiments were combined. Disease severity in Table 1 was calculated by setting the necrotic area of control (untreated) plants as 100% of disease severity, and calculating the percentage of necrotic area reduction with every treatment.

### Nucleic acid isolation and RT-qPCR analysis
Fungal DNA was extracted from leaves using a DNA extraction kit (Qiagen, Hilden, Germany), and the levels of fungal *Elongation factor 1-α* (*MoEF1-α*) or *Glyceraldehyde-3-phosphate dehydrogenase* (*MoGPD*) were quantified by qPCR and normalized using *Bd Ubiquitin-10* (*BdUbiquitin10*) (Suppl. Data 4). RNA was extracted using Direct-zol RNA Purification kit (Zymo Research) according to the manufacturer's instructions. Subsequently, 1 μg of RNA was reversed-transcribed with the RevertAid Reverse Transcriptase cDNA synthesis kit (ThermoFisher) according to the manufacturer's instructions. Quantitative RT-qPCR was performed by using 10 ng of cDNA in the QuantStudio 5 Real-Time PCR system (Applied Biosystems). The QuantStudio software was used to determine the Ct values and transcript amounts were calculated by the $2^{-ΔΔCt}$ method. Primer pairs used for PCR and expression analysis are listed in Suppl. Data 4.

## Production of dsRNA

dsRNA sequences for targeting *GFP*, *SHP*, *MIF1*, and *Pmk1* genes were synthesized with MEGAscript (ThermoFisher) according to the manufacturer's instructions. The *MIF1* and *Pmk1* PCR templates for in vitro transcription were produced from cDNA of axenically grown fungi, using primers harbouring the T7 RNA polymerase promoter sequence (Suppl. Data 4), while templates for GFP-dsRNA and SHP-dsRNA were amplified from pGEM-T-easy plasmids containing *GFP* and *SHP* genes, using primers containing the T7 promoter sequence[41]. For MIF1-ssRNA, the PCR was performed with only the forward primer containing the T7 promoter sequence. Short 21 bp siRNAs (21 bp duplex with overhangs) were purchased from Eurofins (non-methylated) and reconstituted in Milli-Q water. Annealing of the strands was performed by heating up to 90°C the sense and antisense strand (50 µM each) in 5× reaction buffer (ThermoFisher) to a final concentration of 20 µM for 1 min, and cooling down the reaction to 37°C for 45 min. A list of sequences used in the study can be found in Suppl. Data 3, in red font, the sequence used for dsRNA production. Primers for in vitro transcription are shown in Suppl. Data 4. dsRNA constructs were designed and tested for off-target effects with the SiFi software[64].

Long GUS-derived dsRNA sequences (488, 790, 981, 1451 and 1775 bp; Suppl. Data 2) were produced and labeled with the HighYield T7 AF488 RNA Labeling Kit (Jena Bioscience) according to the manufacturer's instruction. A silencer siRNA Labeling Kit (ThermoFisher) with Fluorescein dye was used to label Phi6-dsRNA and 21 bp GAPDH siRNA provided with the kit. For leaf drop experiments, 490 bp dsRNA was labeled with the same kit using Cy3 as a fluorophore.

Phi6-dsRNA was produced in *Pseudomonas syringae* LM2691[65] stably replicating the phage genome[38]. The produced tri-segmented phi6-dsRNA genome, harbouring kanamycin resistance gene insertion in the largest segment, comprises 2948, 4063 and 7599 bp long dsRNA molecules. After overnight culturing of bacteria in liquid cultures, the cells were collected and dsRNA purified by NucleoZol-chloroform extraction, stepwise LiCl precipitation, and ammonium acetate precipitation. circRNA molecules were a gift from Prof. Bindereif, Institute of Biochemistry, JLU Giessen. DNA oligonucleotide templates (Sigma-Aldrich; Suppl. Data 4) were annealed and ssRNA was produced by in vitro transcription (HighScribe T7 high-yield RNA synthesis kit, NEB), including 30 mM Guanosine 5'-monophosphate (GMP, Merck) and RNaseOut (ThermoFisher) in the master mix, incubating the reaction for 2 h at 37°C. Afterwards, DNA template was digested with RQ1DNase (Promega) at 37°C for 30 min. The transcript was purified with the Monarch RNA purification kit (NEB) and quantified with Qubit™ RNA broad-range assay kit (ThermoFisher). The RNA was circularized by using 200 U of T4 RNA ligase (ThermoFisher) in 200 µL of buffer containing 0.1 mg/mL of BSA and RNaseOut (ThermoFisher). Afterwards, circRNA was further purified with phenol/chloroform (Roth) followed by ethanol precipitation. Finally, circRNA was purified from denaturing polyacrylamide gel. Circularization was confirmed by RNase R treatment (Biozym) at 37°C for 25 min and visualized by denaturing polyacrylamide gel stained by ethidium bromide[66].

## Fungal uptake of fluorescent dsRNA

Conidia ($10 \times 10^3$ mL$^{-1}$) were isolated in 0.002% (v/v) Tween20 as described above and 100 µL of fungal suspension were incubated with 10 ng/µL of dsRNA labelled with Fluorescein (21 bp siRNA and Phi6-dsRNA) or Alexa Fluor® 488 dye (rest of dsRNAs) for 24 h at room temperature (RT) in microtiter plates wrapped in aluminium foil. Subsequently, confocal laser scanning microscope (CLSM) analysis was performed for fluorescence accumulation in the GTs.

Treatment of dsRNA with micrococcal nuclease (MNase, Thermo-Fisher) was performed by pelleting down the germinated mycelia and replacing 50 µL of the supernatant with 50 µL of MNase optimal buffer (50 mM Tris-HCl pH 8, 5 mM CaCl$_2$). Afterwards, the mycelia were treated with 381 units of MNase and incubated at 37°C for 30 min.

Uptake from treated leaves was determined by treating intact second youngest leaves of three-week-old *Bd* plants with 20 µL drops (10 ng/µL) of Cy3-labeled dsRNA (490 bp; Suppl. Data 3). The treated areas were drop-inoculated 3 hpt (after dsRNA dried out) with 10 µL drops containing 100 *Mo* conidia. Two additional inoculation points were place ca. 1 cm up and down from the dsRNA application point. CLSM was performed 48 hpi.

## Germ tube elongation assay

GT elongation was assessed by monitoring conidia germination on hydrophobic coverslips (Sigma-Aldrich, Saint Louis, USA). A conidial suspension (200 µL; $5 \times 10^3$ conidia mL$^{-1}$) was incubated in Eppendorf tubes with different dsRNAs (10 ng/µL) for 24 h in a shaker. Subsequently, 30 µL drops of this conidial suspension were placed on coverslips and incubated at RT for additional 24 h. Afterwards the germinating conidia were observed by CSLM and GT length was analysed with ImageJ.

## Microscopy of *Mo*-inoculated *Bd* leaves

Second youngest leaves of 3-week-old *Bd* plants were cut and placed on 1.5% agar plates. 20 µL drops of 10,000 conidia/mL in 0.002% (v/v) Tween20 were placed in randomized locations of the *Bd* leaf as control or together with 10 ng/µL of Pmk1-dsRNA (448 bp). After 2 and 4 dpt, leaves were harvested and stained. Briefly, leaves were de-stained in tri-chloroacetic acid (0.15% w/v) in EtOH-Chloroform (4:1 v/v) solution. Afterwards, leaves were treated with 10% KOH for 1 h at RT in the dark. Then, KOH was removed and leaves were washed with Phosphate Buffer Solution (PBS, pH 7.4). Leaf cuttings were placed 10 µg/mL Alexa Fluor® 488 dye-WGA and 0.05 mg/mL Calcofluor white in PBS (pH 7.4) with 0.02% Silwet L-77 and vacuum infiltrated three times. Samples were left in staining solution overnight at RT in the dark. Images were taken using an Axiovision fluorescence microscope AXIO Imager A2 (Zeiss). AF488-WGA (wheat germ agglutinin) staining was monitored with the Cyan filter [λexcitation (nm): 485 + /- 20; λemission (nm): 515], while for the calcofluor white the UV settings were used [λexcitation (nm): 365; λemission (nm): 420].

## HOG pathway analysis

We tested the activation of the HOG pathway using a *Mo* reporter strain expressing a chimeric MoHOG1::GFP fusion protein under the control of the EF1-α promoter[34]. Conidia (20/µL) in 0.002% (v/v) Tween20 were incubated with 20 ng/µL of the respective RNA. Translocation of the MoHOG1::GFP from the cytosol into the nucleus was monitored in conidia of *Mo* over time. Similar to previous studies, HOG pathway-activating salt stress KCl (0.25 M) was used as a positive stressor control, whereas the antibiotic geneticin was used as a negative control[34].

## Confocal laser scanning microscopy

Images were taken using a Leica TCS SP8 confocal laser scanning microscope (Wetzlar, Germany) equipped with a 75-mW argon/krypton laser (Omnichrome, Chino, CA) and a water immersion objective (HCX APO L40x0.80 W U-V-l objective). Images were processed using the Leica LAS X software. AF488 and GFP [λexcitation (nm): 501; λemission (nm): 591]. Cy3 [λexcitation (nm): 565; λemission (nm): 626].

## ROS detection

*Mo* was propagated on Oatmeal agar (50 mg/L Oatmeal, 10 mg/L agar) and conidia were harvested after two weeks in 0.002% (v/v) Tween20 and filtered using 2 layers of miracloth (Merck KGaA, Germany). Conidia concentration was adjusted to 3500 conidia/mL. *Mo* conidia solution was then incubated in 2.2 mL Eppendorf on a shaker at RT for 48 h. *Mo* mycelia were first immersed in tubes containing different dsRNAs, as well as low molecular weight ( ~ 350–850 bp) poly(I:C), purchased as a potassium salt (Merck KGaA, Germany). After 2 min in dsRNA solution, mycelia were immersed in 2',7'-dichlorodihydrofluorescein diacetate (H$_2$DCFDA) staining solution for an additional 1 min before microscopic analysis. The H$_2$DCFDA staining (Sigma-Aldrich) solution was prepared by resuspending the 19.5 mg in 1 mL of ethanol to make a final solution of 40 mM[32]. Before putting the samples on glass slide, they were

submerged in Milli-Q water for another 1 min. CLSM was performed using the GFP detection as described above. The experiment was repeated 2 times with similar results. Representative pictures of each treatment are shown.

### dsRNA formulation in NPs

dsRNA was encapsulated within dsRNA-alginate-chitosan nanoparticles (NPs)[67]. A 50 ng/μL chitosan solution (30 kDa, 90% deacetylation; Glentham Life Sciences Ltd, United Kingdom) dissolved in 0.005% (v/v) acetic acid, and a 50 ng/μL anionic polymer solution consisting of an equal concentration of dsRNA and of sodium alginate (Algogel 3001, Cargill, USA), were heated to 45°C for 1 min. Subsequently, the chitosan solution was pipetted into the anionic solution at a volume ratio of 1:1.17 (chitosan solution:anionic solution), ensuring a positive to negative charge ratio of 1.25:1, i.e. the ratio of chitosan's positively charged amine groups to negatively charged phosphate and carboxyl groups of dsRNA and alginate, respectively. The resulting mixture was briefly vortexed and incubated at RT for 1 h. Samples were taken for particle characterization and the remaining NPs were stored at 4°C. In addition, alginate-chitosan control NPs were created at the same positive-to-negative charge ratio, therefore, the volume ratio was adjusted to 1:0.94 (chitosan solution:alginate solution). The mean hydrodynamic diameter and size distribution of the NPs was determined by dynamic light scattering at a scattering angle of 165°, and the surface charge was determined by electrophoretic light scattering (Delsa Nano C, Beckman Coulter, USA). The NP size and morphology was also investigated by scanning electron microscopy.

### Statistics and reproducibility

ANOVA test with Pairwise Comparisons with Bonferroni adjustment, ANOVA with post-hoc Tukey HSD test and Dunnett's, Welch's ANOVA or Kruskal Wallis with Dunn's test with Bonferroni adjustment, two-tailed Student's *t*-test or Mann-Whitney were selected after analysis of the sample distribution and homocedasticity in the different groups. The percentage of reduced fungal biomass was assessed with one sample *t*-test. A description of each statistical analysis can be found in figure legends. Replicates are defined as biological replicates, resulting from analysis of n ≥ 3 or pooling 3 independent experiments together for statistical analysis. The sample size is described in each figure legend.

### Reporting summary

Further information on research design is available in the Nature Portfolio Reporting Summary linked to this article.

## Data availability

All data generated or analysed during this study are included in this published article [and its supplementary information files]. Source data underlying the graphs can be found in Supplementary Data 1. All the sequences and primers used in this study are contained in Supplementary Data 2-4.

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

## Acknowledgements
We thank Martina Claar, Christina Birkenstock, Meike Saul-Reiss (IBWF) and Tanja Westerholm for their excellent technical support. We would also like to thank Dr. Jafargholi Imani and Bernhard Werner for their assistance with the microscopy experiments and Prof. Albrecht Bindereif and Silke Schreiner for supplying the circRNA. This work was funded by the Deutsche Forschungsgemeinschaft in the frame of the Research Unit "exRNA" (RU5116) coordinated by KHK. It was also supported by the DFG (grant 459501999 to KHK and AP), the Agence National de la Recherche (ANR grant ANR-21-SUSC-0003-01 to MH), and the Ministry of Agriculture and Forestry Finland (grant VN/8755/2021 to MMP) as part of the project BioProtect coordinated by MH and carried out under the second call of the ERA-NET Cofund Sus-Crop, being part of the Joint Programming Initiative on Agriculture, Food Security and Climate Change (FACCE-JPI). SusCrop has received funding from the European Union's Horizon 2020 research and innovation programme under grant agreement No 771134. In addition, this work was funded by Research Council of Finland grant 331627 to MMP.

## Author contributions
M.L.C. and K.H.K. wrote the manuscript; K.H.K. and M.L.C. designed the study; Y.Z., M.L.C. prepared material for the experiments; M.H. and A.R.S. provided methodology and instrumentation; Y.Z. and S.J. conducted the experiments on *MoHOG1::GFP*; B.M., D.S. and A.P. generated the dsRNA formulation; M.M.P. generated Phi6-dsRNA; C.P. performed fluorescence microscopy and some experiments; M.L.C., K.H.K. and Y.Z. analysed all data and drafted the figures. All authors commented and reviewed the final manuscript.

## Funding

## Competing interests
The authors declare no competing interests.
