## [Transparent Peer Review file · Communications Biology]

Exogenous dsRNA triggers sequence-specific RNAi and fungal stress responses to control *Magnaporthe oryzae* in *Brachypodium distachyon*

Corresponding Author: Dr Maria Ladera-Carmona

Version 0:

Reviewer comments:

Reviewer #1

(Remarks to the Author)

In this work, Zheng et al. show that dsRNA induce fungal stress pathways in addition to RNAi. The stress induced by RNAs is sequence non-specific and occurs (transiently) at high dsRNA concentrations when compared to the RNAi activity. *Magnaporthe oryzae* did take up exogenous dsRNA, and the length of exogenous dsRNA is not critical for fungal uptake. The unspecific effects of dsRNA on Mo, were possibly related to HOG stress pathway activation or ROS. Interestingly, lower exogenous RNA concentrations showed lower immunogenic activity, but the RNAi activity remained. Additionally, the authors generated novel alginate chitosan nanoparticles to be used to deliver dsRNA. dsRNA was released from these nanoparticles was shown to activate RNAi similarly to naked dsRNA. The evidence of sequence-nonspecific PAMP effects occur together with the sequence-specific RNAi activity of exogenous dsRNA is of great interest and has implications for the development and application of dsRNA-based pesticides in agriculture.

The experimental work is accurate, and the experiments well performed. The manuscript is easy to read and enjoyable, describing well the rationale behind each experiment. There is one aspect that in my opinion needs to be addressed: although the author's identified the HOG (and/or ROS) stress pathway activation, there is no explanation for the mechanism by which exogenous RNAs can be recognized by the fungus and whether, besides the HOG/ROS hypothesis, there are other possible explanations for the observed results. Is activation of the HOG pathway sufficient for explaining the enhanced fungal tolerance in plants exposed to Mo? Is the plant somehow contributing to the mechanisms of tolerance in plants treated with exogenous RNAs, either targeting a Mo gene or as an unspecific RNA sequence? The authors claim that the treatment with exogenous RNAs should be considered as a "contact fungicide" with little (if any) role for plant-uptaken exogenous RNAs. It would be interesting to see the unpublished results mentioned in the Discussion ('initial orienting experiments in our lab failed to detect a systemic transport of exogenous dsRNA'). What if it is the plant that perceives exogenous dsRNAs as elicitors of plant defense? While it might be complex to identify the RNA-sensing machinery in Mo that possibly triggers the sequence-unspecific effects, it is worth at least excluding the existence of an RNA-dependent plant induced tolerance to Mo.

Reviewer #2

(Remarks to the Author)

1. Figure 1A and B: It is not clear where the spores were incubated for germination? What kind of surface was used for germination? For Figure 1C, if the dsRNA is dried, how it is expected that the spores would take up the dsRNA? What is the explanation of the uptake of dsRNA?

2. Figure 2A: Germtube is longer than the control. Untreated is not the right kind of control. Control should be everything else excluding dsRNA. What do the authors mean by saying as immunogenic effect is not understood.

3. How do you define non-specific exogenous dsRNA? For Figure S1, why SHP is used? Is GFP not enough?

4. How do you correlate the germtube length? In Figure 2A, germtube length goes significantly up in GFP- dsRNA but in Figure 3, when GFP- dsRNA is used, how the size of necrotic area goes up? Should not it be down?

For the wild type (WT), when you are applying dsRNA or siRNA, how do you get different results in germ tube length and necrotic lesion? There is no correlation. It is not clearly written which genotype is used as control, it is assumed as WT.

5. Figure 4: Not much difference is seen in the movement of GFP-MoHog1p in the given time scale and the scales are different during the given time. So, it is not understandable. How does suddenly a circular RNA come in picture is not clear.

6. The authors claimed that dsRNA concentration at 0.03ng/μl and below, nuclear translocation of chimeric HOG protein was no longer observed. But it is in contradiction to what is seen in Figure 4C where even at a concentration of 0.016ng/μl green dots are clearly visible.

7. It is not clearly described how the authors checked the properties of nanoparticles.

8. In Figure 8C, how does the Pmk1-dsRNA-NP become increasingly effective when inoculation was given after 7 days of spraying dsRNA? Explanation of the longer time interval with increased effectivity of dsRNA-NP is currently missing in the manuscript.

9. In a given system where RNAi is already working as a protective mechanism, is it possible to rule out functioning of PTI, simply by looking at the phenotype? Do the authors have any specific evidence for non-functional PTI?

10. It is observed that same type of non-specific dsRNA molecules have not been used across the experiments. Such an approach would confuse the readers. Moreover, it is difficult to interpret the consistency of the results across the experiments scientifically.

Reviewer #3

(Remarks to the Author)

Here is my review of the manuscript titled "Exogenous dsRNA-Mediated Control of Rice Blast Fungus Involves Sequence-Specific RNAi and Sequence-Unspecific Fungal Stress Responses." I must say it has been a long time since I have enjoyed reading and reviewing dsRNA-related research. It is such a well-detailed, well-structured, and thoroughly executed piece of research. The authors have not only provided deep insights into some of the puzzling observations many researchers, including myself, have encountered with dsRNA applications at the basic science level, but they have also effectively linked these findings to agronomic applications.

While there are sections where I felt a deeper analysis or further discussion could enhance the work, I am reluctant to request major revisions or additional experiments that might delay its publication, as I believe the research community will greatly benefit from it. However, I suggest the authors consider expanding on the following points, either in the introduction or discussion sections:

1. The relationship between germ tube elongation and stress in various pathosystems.
 2. Could non-specific dsRNA serve as a nutrient or nitrogen source for fungal spores, promoting germ tube elongation? If so, would longer dsRNA sequences or higher concentrations enhance this effect? I believe literature exists to support this notion.
 3. In the discussion section, the authors should also address the possibility that the observed reduction in fungal growth, as evidenced by lesion formation and gene silencing, might also be mediated by the host's RNAi machinery in response to dsRNA applications. Recent studies may provide support for this mechanism.
- Overall, this is an excellent piece of research, and I believe it is highly deserving of publication in a prestigious journal such as CB.

Version 1:

Reviewer comments:

Reviewer #1

(Remarks to the Author)

Although the authors failed to address most of my concerns experimentally, I believe this work is of interest. The revised version is certainly improved and I do not have additional comments.

Reviewer #2

(Remarks to the Author)

The authors of the manuscript "Exogenous dsRNA triggers sequence-specific RNAi and broad fungal stress responses to control Rice Blast fungus" have addressed all the concerns raised and incorporated relevant details in the said manuscript. Therefore, you may please go ahead with the publication process of the same.

Reviewer #3

(Remarks to the Author)

I am highly satisfied with the responses to my comments and strongly recommend the publication of this manuscript. Many dsRNA researchers face challenges due to unusual and unexpected results observed upon exposure to specific and non-specific dsRNA. This article provides valuable insights that help explain at least some of these observations.

Point to point response to reviewers (COMMSBIO-24-3604-A)

We appreciate the effort to help improving the MS. We considered all key points raised in our point-to-point response to the reviewers. In general, we have extended some points raised by the reviewers in our introduction or discussion; included an additional experiment (Fig. 10); changed the axis in Figure 2 and 8; change figure order to make our message straighter forward; improved the quality of our figures, and transformed the bar graphs into dot blot to show data distribution.

Reviewer #1 (Remarks to the Author):

....The experimental work is accurate, and the experiments well performed. The manuscript is easy to read and enjoyable, describing well the rationale behind each experiment. There is one aspect that in my opinion needs to be addressed: although the author's identified the HOG (and/or ROS) stress pathway activation, there is no explanation for the mechanism by which exogenous RNAs can be recognized by the fungus and whether, besides the HOG/ROS hypothesis, there are other possible explanations for the observed results.

We appreciate this statement of rev1. To take up this point we extended our discussion on the possible mode of action of dsRNA perception in fungi by comparing it with the perception systems in animals and plants. In conclusion, there is no dsRNA receptor described so far in fungi and further research is required to gather more information on it. Lines 403-408 "We hypothesized that ROS burst triggered by dsRNA (Fig. 3) impacted germ tube development (Fig. 2) and Hog1p migration to the nucleus (Fig. 4). Microscopic analysis of *in planta* dsRNA treatments and *Mo* infection showed that dsRNA-treated conidia produced more aerial hyphae and reduced plant penetration (Fig. S3). Exogenous dsRNA perception might take place by any of the components upstream the MoHOG1p signalling cascade, responding to osmotic stress but also to dsRNA."

Rev1: Is activation of the HOG pathway sufficient for explaining the enhanced fungal tolerance in plants exposed to Mo?

We have expanded the discussion on dsRNA-mediated HOG1 activation in the MS. But we have not shown in our experiments an enhanced fungal tolerance upon dsRNA applications. We just tested the HOG1 pathway as a proxy for a fungal stress response.

The HOG1 signalling pathway is essential for the survival and adaptability of fungi, as has been shown in many publications (e.g., Bohnert et al. 2019). By controlling responses to osmotic and other environmental stresses, it ensures that fungi can thrive in diverse and often challenging habitats.

In the present work, we add to this knowledge that dsRNA - apart from its known RNAi effect - also induces the signalling pathway. The signalling and responses of the HOG1 pathway are extremely complex, have many facets and also depend on the fungal microbe; accordingly, we cannot answer the question what consequences the activation of the HOG1 pathway would have for the fungus without opening up a completely new avenue of research. Instead, the HOG1 signalling pathway experiments simply provided us with a very reliable and sensitive readout of the antifungal activity of dsRNA, which has not been described before.

Rev1: Is the plant somehow contributing to the mechanisms of tolerance in plants treated with exogenous RNAs, either targeting a Mo gene or as an unspecific RNA sequence?

If we understand this question correctly, rev1 is asking about the PTI effect of dsRNA on the host plant. Several publications show that dsRNA triggers a PTI response in plants, thus enhancing the disease

resistance (in fact, most critical ones are from our labs: Niehl et al. 2016; Huang et al. 2022). Although, notably, a PTI response to dsRNA in monocyledonous plants such as *Brachypodium distachyon* has not been reported.

We have already discussed this point in our MS, but in the light of the rev1's question, we have expanded on it in the new version. See lines 425-427. Overall, this well-known phenomenon is beyond the scope of our manuscript. We focus on the new aspect of fungal responses to dsRNA. We have outlined this in the Introduction and Discussion, and in particular addressed the question of whether the plant response contributes to higher dsRNA-induced resistance. We explicitly leave it to the editor's decision to add further information (experiments and figures) on the PTI response of our host plant, *Brachypodium*, to dsRNA, but we feel that this aspect dilutes the focus of our manuscript.

Rev1: The authors claim that the treatment with exogenous RNAs should be considered as a “contact fungicide” with little (if any) role for plant-uptaken exogenous RNAs. It would be interesting to see the unpublished results mentioned in the Discussion (‘initial orienting experiments in our lab failed to detect a systemic transport of exogenous dsRNA’). What if it is the plant that perceives exogenous dsRNAs as elicitors of plant defense? While it might be complex to identify the RNA-sensing machinery in Mo that possibly triggers the sequence-unspecific effects, it is worth at least excluding the existence of an RNA-dependent plant induced tolerance to Mo.

The reviewer is correct in this consideration. But see our response above. The uptake of dsRNA into plants is subject of many papers. Yet, it is still controversially seen in the literature. As a general notion, the uptake of naked exogenous dsRNA into plant leaves is very weak and most papers argue it remains apoplastic. **As requested by the rev1 we have now extended this aspect by conducting additional experiments (show in Fig 10).** We also extended our discussion on dsRNA-triggered PTI as indicated above.

Reviewer #2 (Remarks to the Author):

Figure 1A and B: It is not clear where the spores were incubated for germination? What kind of surface was used for germination?

The details on this experiment were given in the MM chapter. We revised the legend to the figures to make it more convenient for the reader. “Conidia ... were isolated in 0.002% (v/v) Tween20 ... and 100 μ L of fungal suspension were incubated with 10 ng/ μ L of dsRNA labelled with Fluorescein or Alexa Fluor® dsRNAs for 24 h at room temperature (RT) in microtiter plates wrapped in aluminium foil.”

Rev2: For Figure 1C, if the dsRNA is dried, how is it expected that the spores would take up the dsRNA? What is the explanation of the uptake of dsRNA?

We revised the MS to exclude any misunderstanding. Even though we waited 3 h for the dsRNA to settle on the leaf, we treated the same spot with a drop containing *Mo* conidia, where the dsRNA could have been resuspended. Despite this, there is always a “wet” micro-environment on the leaves under the hyphae. Fungi take up a lot of metabolites from the leaf surface.

Similarly to what was found in *Sclerotinia sclerotiorum* (Wytinck et al. 2020), growing hyphae have thinner cell wall allowing exogenous dsRNA uptake (probably via clathrin-mediated endocytosis). Conidia have thicker cell walls and cannot hardly take up dsRNA. Indeed, we found fluorescent signal in the germinating germ tube, not in the conidia (as Fig. 1 in liquid culture). We have expanded our discussion in this direction (lines 348-350).

Rev2: Figure 2A: Germ tube is longer than the control. Untreated is not the right kind of control. Control should be everything else excluding dsRNA.

In this experiment, we wanted to specifically test the effect of sequence-specific dsRNA vs. control sequences (unspecific sequences = sequences that have no target in the fungus). We tested all the sequences we had in our hands as long and short duplexes, in addition to the synthetic analogue poly(I:C), which is commonly used in mammal studies and plants (Niehl et al. 2016). Additionally, we tested ssRNA sequences. We believe that the outcome of this experiment was enough to continue our investigations.

Rev2: What do the authors mean by saying as immunogenic effect is not understood.

We are sorry for the confusion. We introduced this topic in the Introduction lines 69-75. In mammalian cells, dsRNA has an immunogenic effect, triggering antiviral responses. Similarly, dsRNA triggers PTI responses in plant cells (such as MAPK activation, ethylene production and root length inhibition; (Huang et al. 2022)). Whether dsRNA affects fungi in a similar manner was not known.

Rev2: How do you define non-specific exogenous dsRNA? For Figure S1, why SHP is used? Is GFP not enough?

The term “non-specific dsRNA” was used for all dsRNA with no target in the receiving organism as calculated by the SiFi software based on the target organism’s global transcriptome sequence. In our study we tested up to four non-specific dsRNAs with confirmed lack of target sequences in *M. oryzae*, i.e. dsRNA-SHP derived from the *Sitobion avenae* (grain aphid) *Sheath Protein (SHP)* gene, dsRNA-GFP derived from the *GFP* gene, poly(I:C), and the phage phi6 dsRNA produced in high amounts in a bacterial system. We strongly believe that the use of non-specific dsRNA is essential in our study, and we use different dsRNA to increase the reliability and power of our experiments.

Rev2: How do you correlate the germ tube length?

In this experiment, pictures are taken with a confocal microscope, exported with a scale bar of known distance for further ImageJ analysis, where the size of the scale bar is set as reference to measure all the germ tube lengths. Then all the individual measures are plot in the graph and statistical analysis is performed. We hope we were able to clarify this question.

Rev2: In Figure 2A, germ tube length goes significantly up in GFP- dsRNA but in Figure 3, when GFP- dsRNA is used, how the size of necrotic area goes up? Should not it be down?

Yes, it is down. In Figure 3 (now Fig. 5) the size of necrotic area is reduced when GFP-dsRNA or GFP-siRNAs were sprayed (see the median in the box plot). Of note, these two different experiments were carried out in different concentrations: germ tube length (Figure 2) at 10 ng/μL and spray assay (old Figure 3 and now Fig. 5) at 1 ng/μL. In Figure S2 we performed in planta spray assay with 10 ng /μL and all the dsRNA sprayed significantly reduced fungal infection. Abnormally elongated GT due to dsRNA treatments correlates with reduction of necrotic area.

Rev2: For the wild type (WT), when you are applying dsRNA or siRNA, how do you get different results in germ tube length and necrotic lesion? There is no correlation. It is not clearly written which genotype is used as control, it is assumed as WT.

As stated in the MM, we always used the WT strain (70-15). For the experiment shown in Fig. 2B, we additionally used the *M. oryzae* mutant $\Delta Momifl-1$. In this case, correlation between germ tube length and necrotic lesion when using dsRNA or siRNAs is difficult to make since the experiments were performed at different concentrations, *in vitro* for GT length and *in planta* for spray assays.

Rev2: Figure 4: Not much difference is seen in the movement of GFP-MoHog1p in the given time scale and the scales are different during the given time. So, it is not understandable.

We tried to stick to the time points investigated in Bohnert et al. 2019, where the positive stressor KCl should transiently induce nuclear transfer of the GFP signal within 5 minutes, which is then quickly reversible at 60 min. This time points are widely accepted to test the transfer of the HOG1 protein, since Dr. Stefan Jacob (co-author in this manuscript) previously tested several time points for HOG activity and chose 5 min and 60 min as the most representative ones (see Bohnert et al. 2019). To avoid misunderstanding, we have improved the microscope pictures shown in Figure 4. Thank you for pointing this out.

Rev2: How does suddenly a circular RNA come in picture is not clear.

The reviewer is right: the use of circRNA needs more explanation. In the HOG1 experiment we wanted to test several different types of RNA envisaged to be used in future plant protection to explore if they exploit an immunogenic effect on *M. oryzae*. Hence, we also tested circRNA which is a continuous loop with no 5' or 3' end. In mammal cells, circRNAs evade immune responses. We wanted to test whether this is the case in fungal cells - and could confirm it.

Rev2: The authors claimed that dsRNA concentration at 0.03ng/μl and below, nuclear translocation of chimeric HOG protein was no longer observed. But it is in contradiction to what is seen in Figure 4C where even at a concentration of 0.016ng/μl green dots are clearly visible.

At 0.016 ng/μL no nuclear GFP signal was observed. Instead, at 0.032 ng/μL only 4 out of 13 conidia investigated at the two time points showed nuclear accumulation. We changed Fig. 4 to avoid confusion. This concentration was chosen for further experiments to have some effect. Of note, plant protection at this concentration is sequence-specific but not as strong as before.

Rev2: It is not clearly described how the authors checked the properties of nanoparticles.

We have added additional information to MM section: "The mean hydrodynamic diameter and size distribution of the NPs was determined by dynamic light scattering at a scattering angle of 165°, and the

surface charge was determined by electrophoretic light scattering (Delsa Nano C, Beckman Coulter, USA). The NP size and morphology was also investigated by scanning electron microscopy.”

Rev2: In Figure 8C, how does the Pmk1-dsRNA-NP become increasingly effective when inoculation was given after 7 days of spraying dsRNA? Explanation of the longer time interval with increased effectivity of dsRNA-NP is currently missing in the manuscript.

Thank you for this question. There are several examples in the literature that the advantage of NP formulations is increasing with the gap time. The explanation for this is that during this period the naked dsRNA was degraded more and more, reducing its effectiveness. Pmk1 dsRNA formulated in NP was stabilised and had a greater effect in reducing *Mo* growth. Lines 290-293 “Hence, non-formulated *Mo*-specific dsRNA could protect *Bd* plants from *Mo* infection to a certain extent, but this protective effect is prolonged when the dsRNA is protected with the alginate-chitosan formulation.”

Rev2: In a given system where RNAi is already working as a protective mechanism, is it possible to rule out functioning of PTI, simply by looking at the phenotype? Do the authors have any specific evidence for non-functional PTI?

We have tried to address this matter expanding our discussion. See comments above about plant PTI.

Rev2: It is observed that same type of non-specific dsRNA molecules have not been used across the experiments. Such an approach would confuse the readers. Moreover, it is difficult to interpret the consistency of the results across the experiments scientifically.

Yes, thank you to clarify this in the manuscript. The use of different non-specific dsRNA with very different RNA sequences in our view is very important to support of our hypothesis that non-specific dsRNA has effects on *M. oryzae*. We did not want to rely on a single type of unspecific dsRNA, therefore we used various of them. However, the reviewer is right that this needs more explanations. We tried to address this issue in our new version of the manuscript.

Reviewer #3 (Remarks to the Author):

" I must say it has been a long time since I have enjoyed reading and reviewing dsRNA-related research. It is such a well-detailed, well-structured, and thoroughly executed piece of research. The authors have not only provided deep insights into some of the puzzling observations many researchers, including myself, have encountered with dsRNA applications at the basic science level, but they have also effectively linked these findings to agronomic applications.

While there are sections where I felt a deeper analysis or further discussion could enhance the work, I am reluctant to request major revisions or additional experiments that might delay its publication, as I believe the research community will greatly benefit from it. However, I suggest the authors consider expanding on the following points, either in the introduction or discussion sections:

1. The relationship between germ tube elongation and stress in various pathosystems.

Thank you very much for your comments, your feedback is really appreciated.

Concerning your question, Yin et al. 2016 found that nitric oxide and ROS coordinate germination and germ tube length in *Puccinia striiformis*. H₂DCFDA staining revealed ROS accumulation in the germinating hyphae. Based on this finding, we hypothesized that ROS burst triggered by dsRNA (old Figure 5-new Fig. 3) caused increased germ tube in treated conidia (Figure 2) and Hog1p migration to the nucleus (Figure 4). Microscopic analysis of in planta dsRNA treatments and *Mo* infection, showed

that dsRNA-treated conidia produced more aerial hyphae and reduced plant penetration (old Figure S2 now Fig. S3). Reduced penetration would lead to reduced necrotic area in spray assays. We have addressed this hypothesis in the discussion. We also changed the figure order to clarify this. Line 390-391: "...while ROS is needed for proper GT elongation in *Puccinia striiformis*²⁷ where H₂DCFDA staining revealed ROS accumulation in the germinating hyphae". Lines 403-407: "We hypothesized that ROS burst triggered by dsRNA (Fig. 3) impacted germ tube development (Fig. 2) and Hog1p migration to the nucleus (Fig. 4). Microscopic analysis of in planta dsRNA treatments and *Mo* infection showed that dsRNA-treated conidia produced more aerial hyphae and reduced plant penetration (Fig. S3)."

Rev3: Could non-specific dsRNA serve as a nutrient or nitrogen source for fungal spores, promoting germ tube elongation? If so, would longer dsRNA sequences or higher concentrations enhance this effect? I believe literature exists to support this notion.

Thank you for this question. We included discussion on this point to the discussion. We found similar results by using dsRNA or siRNAs in the same amount (ng), and a pure nutrition mechanism does not fit to our data related to ROS and HOG1 not to the resistance data on *Mo*.

Rev3: In the discussion section, the authors should also address the possibility that the observed reduction in fungal growth, as evidenced by lesion formation and gene silencing, might also be mediated by the host's RNAi machinery in response to dsRNA applications. Recent studies may provide support for this mechanism.

Overall, this is an excellent piece of research, and I believe it is highly deserving of publication in a prestigious journal such as CB.

We appreciate very much the rev3's view. He is right that we need to have an additional note on the PTI inducing effect of dsRNA in *Brachypodium*. We included it in the discussion in our new version of the MS. Lines 425-429: "It is well known that exogenous dsRNA induces canonical PTI responses in plants, leading to enhanced protection against viral infection⁹. Therefore, while do not rule out dsRNA-triggered PTI in Bd plants, we have focused on dsRNA-triggered nonspecific effects in the fungus, which are less well studied. Either way, both mechanisms would contribute to dsRNA-mediated protection against phytopathogenic fungi".